# Feature Importance Metrics in the Presence of Missing Data

**Henrik von Kleist** [1 2 3]   **Joshua Wendland** [4]   **Ilya Shpitser** [3]   **Carsten Marr** [1]

## Abstract

Feature importance metrics are critical for interpreting machine learning models and understanding the relevance of individual features. However, real-world data often exhibit missingness, thereby complicating how feature importance should be evaluated. We introduce the distinction between two evaluation frameworks under missing data: (1) feature importance under the full data, as if every feature had been fully measured, and (2) feature importance under the observed data, where missingness is governed by the current measurement policy. While the full data perspective offers insights into the data generating process, it often relies on unrealistic assumptions and cannot guide decisions when missingness persists at model deployment. Since neither framework directly informs improvements in data collection, we additionally introduce the feature measurement importance gradient (FMIG), a novel, model-agnostic metric that identifies features that should be measured more frequently to enhance predictive performance. Using synthetic data, we illustrate key differences between these metrics and the risks of conflating them.

## 1. Introduction

Feature importance metrics play a crucial role in supervised machine learning, offering a means to evaluate the impact of individual features on a model's performance. A subset of these metrics, known as global feature importance metrics (Ewald et al., 2024), determine the value of including a feature in the prediction model, typically given a certain baseline of other features. These metrics help clarify the relevance of features in applications like medical decision-making, where they can provide insights into the diagnostic and prognostic value of medical tests (Yoo et al., 2025; Rai et al., 2024; Ma et al., 2023). In practice, however, particularly in clinical contexts, features are often measured inconsistently, leading to missing values (Zamanian et al., 2024). This raises important questions about how to define and interpret feature importance in such scenarios.

To ground our discussion, we present three simplified scenarios in heart attack diagnosis, each illustrating a distinct perspective on how to reason about feature importance in the presence of missing data:

***Scenario 1:*** A biomedical researcher aims to understand the intrinsic relationship between troponin levels, a biomarker typically measured via a blood test, and the occurrence of heart attacks - independent of hospital testing protocols. Using a retrospective dataset with missing values, the researcher is interested in feature importance as if every feature had been fully measured.

***Scenario 2:*** A machine learning engineer develops a heart attack prediction model and evaluates the relevance of troponin levels. Given a retrospective dataset with missingness, the model is to be trained for deployment in a setting where such missingness will persist due to unchanged hospital testing protocols. The engineer is thus interested in **feature importance under the current missingness** where features sometimes assume a special value "?" representing missingness.

***Scenario 3:*** Since troponin tests are costly and time-consuming, but provide valuable diagnostic information, an analyst seeks to optimize their use after deploying the prediction model. The analyst wants to understand how prediction performance would change if troponin was measured more frequently. They are thus interested in **feature importance under an increase in measurement probability**.

In this work, we examine how feature importance metrics can be defined and estimated in such scenarios. To clarify the structure of the problem, we organize our analysis

[1]Institute of AI for Health, Helmholtz Munich - German Research Center for Environmental Health, Neuherberg, Germany [2]TUM School of Computation, Information and Technology, Technical University of Munich, Garching, Germany [3] Department of Computer Science, Johns Hopkins University, Baltimore, MD, USA [4] Faculty of Computer Science, Ruhr University Bochum Bochum, Germany . Correspondence to: Henrik von Kleist <henrik.kleist@tum.de>.

*Proceedings of the $42^{nd}$ International Conference on Machine Learning*, Vancouver, Canada. PMLR 267, 2025. Copyright 2025 by the author(s).

around the classical three-step framework of statistical inference: (i) **Estimand definition**, which entails specifying the target parameter to be computed; (ii) **Identification**, which assesses whether the estimand can be expressed as a function of the observed data; and (iii) **Estimation**, which concerns how the estimand is calculated from the observed data.

We find that current practices for reporting feature importance metrics in scenarios with missing data lack rigor across all three steps:

- *Estimands are not clearly specified:* When transitioning from a fully observed dataset to one with missing data, the target parameter - i.e. the feature importance metric - must be redefined to reflect the data structure. In current practice, analysts do not report based on which data the feature importance is calculated. For example, is the metric intended to reflect the importance of features in the observed data with missingness, or in the full data that would have been available had there been no missingness?

- *Identification assumptions are not stated, and identification is not performed:* Analysts often fail to specify the necessary assumptions required for identification. This includes independence assumptions about the missingness process, such as whether data are assumed to be Missing Completely at Random (MCAR), Missing at Random (MAR), or Missing Not at Random (MNAR) (Bhattacharya et al., 2020; Nabi et al., 2020; Mohan & Pearl, 2021). It also includes the positivity assumption (Petersen et al., 2012), which ensures sufficient overlap in feature availability across subpopulations. As a result, it remains unclear under which assumptions the reported feature importance metrics hold true. Examples of this issue can be found in Yoo et al. (2025); Rai et al. (2024); Ma et al. (2023); Beld et al. (2024); Peng et al. (2023); Guan et al. (2023); Zhou et al. (2023); Qi et al. (2022); Xie et al. (2023); Bao et al. (2023); Yang et al. (2022); Zhang et al. (2023).

- *Estimation methods are often biased:* A variety of methods are used in practice to handle missing data when computing feature importance metrics, including but not limited to mean/median/zero imputation (Zhuang et al., 2023; Lucini et al., 2023; Ishii et al., 2023; Huang et al., 2023; Danilatou et al., 2022), conditional mean imputation (Vo et al., 2024), complete case analysis (Zhang et al., 2022), and multiple imputation (Peng et al., 2023; Guan et al., 2023; Qi et al., 2022; Xie et al., 2023; Chen et al., 2023; Zeng et al., 2023). It is often unclear to which estimand these methods correspond to and whether they introduce bias. For instance, some methods implicitly assume that the missing fea-

ture contains no or excessive information, leading to distorted feature importance estimates. Consequently, reported metrics may misrepresent the true underlying importance of features.

To address the lack of estimand specification, we introduce the distinction between two concepts: **(1) Full data feature importance metrics**, which refer to the ground truth data had there been no missingness, and **(2) Observed data feature importance metrics**, which refer to the observed data and thus analyze feature importance in the context of the current measurement policy. We illustrate the implications of these two perspectives using a simple yet practical feature importance metric: the leave-one-covariate-out (LOCO) metric (Lei et al., 2018).

We then turn to the problem of identification. While observed data feature importance metrics are directly identifiable from the observed data distribution, the identification of full data feature importance metrics relies on strong - and often unrealistic - assumptions. In particular, we examine the role of the positivity assumption, which can critically limit the reliability of estimation in practice.

Next, we address the problem of estimation by reviewing commonly used approaches for computing feature importance metrics under missing data. We classify these methods according to the estimands they target and assess their ability to produce unbiased results under various missingness mechanisms. We demonstrate that estimation results can differ drastically between full data and observed data feature importance metrics. For instance, consider a costly or invasive medical test that provides a highly informative feature: if performed for every patient, the feature would be an excellent predictor, yielding high full data feature importance. However, due to cost constraints, it is rarely conducted in practice - leading to a much lower observed data feature importance.

While full data and observed data feature importance metrics are appropriate for the questions posed in Scenarios 1 and 2, they fall short in addressing Scenario 3, where the goal is to assess a features importance under changes in the measurement process. To fill this gap, we introduce the **feature measurement importance gradient (FMIG)**, a novel, model-agnostic, and computationally efficient estimand. FMIG quantifies the sensitivity of prediction performance to small increases in the measurement probability of a given feature. Unlike full data feature importance metrics, FMIG does not require a positivity assumption, making it more robust and practically applicable in real-world settings with high rates of missingness.

## 2. Related Methods

In Appendix A, we provide an overview of feature importance metrics. In the following, we discuss related approaches for assessing feature importance and the closely related problem of feature selection in the presence of missing data.

### 2.1. Estimand Specification

To the best of our knowledge, no prior studies have explicitly differentiated between observed and full data feature importance metrics. However, some papers incidentally report both types of metrics because they use multiple methods - some of which align with observed data feature importance metrics and others with full data feature importance - without noticing the difference in the underlying estimands (Vo et al., 2024; Srinivasan et al., 2016; Rai et al., 2024). The same issue arises in feature selection (Fan et al., 2023; Mera-Gaona et al., 2021; Ergul Aydin & Kamisli Ozturk, 2024). Notably, only Hapfelmeier (2012) acknowledged that multiple imputation addresses a different question than their proposed method, but the difference in estimands remained unexplored. In contrast, our work is the first to formally define these concepts and clarify their implications.

### 2.2. Identification

Challenges related to estimating feature importance or performing feature selection under different missingness mechanisms, such as MCAR, MAR, and MNAR, have been discussed in some studies (Vo et al., 2024; Hapfelmeier, 2012; Srinivasan et al., 2016; Fan et al., 2023; Beld et al., 2024; Zhao & Long, 2017), but MNAR remains particularly underexplored (Zhao & Long, 2017). Many papers fail to specify the missingness assumptions under which their reported metrics are valid (Yoo et al., 2025; Rai et al., 2024; Ma et al., 2023; Beld et al., 2024; Peng et al., 2023; Guan et al., 2023; Zhou et al., 2023; Qi et al., 2022; Xie et al., 2023; Bao et al., 2023; Yang et al., 2022; Zhang et al., 2023). Moreover, no prior work has, to the best of our knowledge, addressed the role of the positivity assumption in the context of feature importance metrics under missing data.

### 2.3. Estimation

Various studies have investigated methods for estimating feature importance metrics (Vo et al., 2024; Srinivasan et al., 2016) or conducting feature selection (Gunn et al., 2023; Seijo-Pardo et al., 2018; 2019; Mera-Gaona et al., 2021; Hu et al., 2021; Genossar et al., 2024; Ergul Aydin & Kamisli Ozturk, 2024; Doquire & Verleysen, 2012) in the presence of missing data. These approaches include imputation-based methods (Zhao & Long, 2017; Vo et al., 2024; Seijo-Pardo et al., 2018; 2019) and methods that in-

trinsically handle missing data (Vo et al., 2024). Among imputation-based methods, conditional mean imputation and similar approaches like KNN-imputation (Smit et al., 2022) or SVD imputation are commonly used (Srinivasan et al., 2016; Hapfelmeier, 2012; Ergul Aydin & Kamisli Ozturk, 2024; Smit et al., 2022), but introduce bias (Seijo-Pardo et al., 2018).

In contrast, multiple imputation (MI) is recognized as an unbiased method for estimating feature importance under missing data (Moons et al., 2006). However, prior work has emphasized that the label must be included in the imputation model to ensure validity (Moons et al., 2006). Despite this recommendation, it is still common in practice to exclude the label (Srinivasan et al., 2016; Hapfelmeier, 2012; Ergul Aydin & Kamisli Ozturk, 2024; Zeng et al., 2023).

While these methods have been analyzed in various studies, no work has systematically categorized them based on the type of feature importance metric they compute - whether observed data or full data feature importance. In addition to this categorization, we identify which estimation methods introduce bias, further clarifying their limitations and appropriate applications.

### 2.4. Action Importance Metrics in Reinforcement Learning and Causal Inference

The feature measurement importance gradient (FMIG), introduced in this work, quantifies how predictive performance would change if the probability of measuring a given feature were slightly increased. This metric frames the measurement process as a decision-making problem, where the decision is whether to measure a feature in a specific context defined by the already-measured features. This perspective naturally aligns with concepts in reinforcement learning (RL) and causal inference, which study the importance of actions and their effects.

Shapley values, which are widely used in feature importance evaluation (Lundberg & Lee, 2017), have been also applied in RL to assess the importance of individual actions (Beechey et al., 2023). Concepts like the credit assignment problem in RL (Pignatelli et al., 2024), which focus on attributing rewards to specific actions, highlight related challenges in quantifying the importance of actions and suggest potential connections to feature measurement tasks.

FMIG also shares a conceptual link with incremental propensity score interventions introduced by Kennedy (2019), which provide an interpretable framework for analyzing the effects of small changes in action probabilities (propensity scores). We adapt this idea to the setting of feature measurement and, by focusing only on the gradient, simplify estimation while retaining an interpretable metric.

# 3. Full Data and Observed Data Feature Importance Metrics

In this section, we introduce notation, define the full data and observed data LOCO metrics, and discuss identification and estimation. We consider a classification setting, but our results naturally extend to regression.

## 3.1. Notation

We define the key variables and functions as follows:

- **Observed features**: $X \in (\mathbb{R} \cup \{"?"\})^d$, where $X_i$ represents the value of the $i$-th feature. If $X_i = "?"$, the feature is missing; otherwise, $X_i \in \mathbb{R}$.

- **Missingness indicators**: $R \in \{0, 1\}^d$, where $R_i = 1$ indicates that feature $X_i$ is observed, and $R_i = 0$ indicates it is missing.

- **Ground truth features**: $X_{(1)} \in \mathbb{R}^d$, representing the true values of all features, had they all been observed. Using potential outcome notation (Rubin, 2005), $X_{(1),i}$ denotes the value of feature $X_i$, potentially contrary to fact, had $R_i = 1$. We obtain:

$$X_i = \begin{cases} X_{(1),i} & \text{if } R_i = 1, \\ "?" & \text{if } R_i = 0. \end{cases} \quad (1)$$

- **Label**: $Y \in \{0, 1, \ldots, K-1\}$, the categorical outcome associated with the feature set $X$, where $K$ is the number of possible classes. We do not consider label missingness in this work.

- **Missingness process**: $\pi \colon \mathbb{R}^d \times \{0, 1, \ldots, K-1\} \to [0, 1]^{2^d}$, representing the mechanism that determines the missingness indicator $R$. In the most general case $\pi(R|X_{(1)}, Y)$ depends on both the ground truth features $X_{(1)}$ and the label $Y$.

- **Classifier**: $f_{cl} \colon (\mathbb{R} \cup \{"?"\})^d \times \{0, 1\}^d \to [0, 1]^K$, defines a mapping from the observed features and missingness indicators to the label probabilities. While $X$ already encodes missingness through the presence of "?" values, we explicitly include the missingness indicators $R$ in the notation to emphasize the dependence on the missingness pattern. Using missingness indicators in the classifier is common practice in many settings (Van Ness et al., 2023; Singh et al., 2021).

- **Loss function**: $L \colon \{0, 1, \ldots, K-1\} \times [0, 1]^K \to \mathbb{R}$, defines a loss function, for example cross entropy, for the classifier $f_{cl}$ based on the true labels and the predicted label probabilities.

We further let $X_{-j}$, $R_{-j}$, $X_{(1),-j}$ denote the reduced dimension observed features, missingness indicators and ground truth features when feature $j$ is excluded. Similarly, we let $f_{cl,-j} \colon (\mathbb{R} \cup \{"?"\})^{d-1} \times \{0, 1\}^{d-1} \to [0, 1]^K$ denote the classifier that excludes information from feature $j$.

## 3.2. Estimand Definition: Full Data and Observed Data LOCO Metrics

For full data, the LOCO metric for feature $j$ evaluates as follows.

*Full data LOCO:*

$$\begin{aligned} LOCO_j^{FD} = & \mathbb{E}\left[L\left(Y, f_{cl,-j}(X_{(1),-j}, R_{-j} = \vec{1})\right)\right] \\ & - \mathbb{E}\left[L\left(Y, f_{cl}(X_{(1)}, R = \vec{1})\right)\right] \end{aligned} \quad (2)$$

Here, the full data LOCO metric measures the expected loss difference if feature $j$ is excluded compared to if it is included, with all other features fully observed.

In contrast, the observed data LOCO metric evaluates as follows.

*Observed data LOCO:*

$$LOCO_j^{OD} = \mathbb{E}\left[L\left(Y, f_{cl,-j}(X_{-j}, R_{-j})\right)\right] - \mathbb{E}\left[L\left(Y, f_{cl}(X, R)\right)\right] \quad (3)$$

It differs from $LOCO_j^{FD}$ in that it replaces the ground truth features $X_{(1)}$ with their observed proxies $X$ and the missingness indicators $R$, which embeds the feature importance in the context of the measurement policy that was used in the acquisition of the dataset. The observed data LOCO metric $LOCO_j^{OD}$ thus quantifies the loss difference when feature $j$ is excluded compared to if it is included, but only when measured under the current measurement policy, with other features partially observed under their respective measurement policies.

*Remark* 3.1. Note that the observed data LOCO metric, $LOCO_j^{OD}$, does not capture the contrast in loss that would occur if feature $j$ were no longer measured. We introduce an alternative estimand, termed *leave-one-covariate-unmeasured (LOCU)*, which describes this scenario in detail in Appendix B. The LOCU metric coincides with $LOCO^{OD}$ only under certain restrictive independence assumptions about the missingness process.

## 3.3. Identification

The observed data LOCO metric is already identified, meaning it is expressed as a function of the observed data. This is not the case for the full data LOCO metric, which depends on the unobserved ground truth features $X_{(1)}$. To identify the full data LOCO metric, missing data identification methods can be applied. Generally, the following assumptions are required:

*Stable Unit Treatment Value Assumption (SUTVA):* This assumption ensures that the counterfactual feature values are well-defined, as described by Eq. 1. Specifically, it requires that the observed features are equal to the counterfactual

features when $R = 1$. Additionally, it assumes that data points do not interfere with one another - specifically, that missingness in one instance is not influenced by the features or missingness indicators of other instances.

***Independence assumptions about the missingness process:*** These assumptions describe the relationship between the missingness process $\pi$ and the data:

- Missing Completely at Random (MCAR): The missingness process is independent of both observed and unobserved data, $\pi(R|X_{(1)}, Y) = \pi(R)$.
- Missing at Random (MAR): The missingness process depends only on the observed data.
- Missing Not at Random (MNAR): The missingness process may depend on unobserved data, such as the ground truth features $X_{(1)}$.

***Positivity assumption:*** The positivity assumption requires that $\pi(R = \vec{1} \mid X_{(1)}, Y) > 0$ for all $X_{(1)}, Y$ with $p(X_{(1)}, Y) > 0$. In essence, it ensures that for every ground truth data point, there is a positive probability that it is fully observed. The positivity assumption is particularly critical in many real-world settings, as it is often violated. Consider, for example, a time-series medical prediction setting. The assumption requires that for every patient, there is a chance that all medical tests are being performed jointly and at every time point - a scenario that is rarely, if ever, met in practice. As a result, full data feature importance metric estimates become highly unreliable in such contexts.

### 3.4. Estimation

In the following, we categorize various estimation methods for both full data and observed data feature importance metrics. Note that feature importance is defined relative to a given classifier $f_{cl}$, and thus reflects the true importance of a feature only to the extent that the classifier accurately captures the relationship between $X$ and $Y$.

#### 3.4.1. Estimation of the Full Data LOCO Metric

Inverse probability weighting (Seaman & White, 2013) and multiple imputation (Sterne et al., 2009) are widely used methods from the missing data literature for estimating full data target parameters. They can thus also be used to compute the full data LOCO metric. Both approaches reconstruct samples from the ground truth distribution $p(X_{(1)}, Y)$, enabling the estimation of full data LOCO metrics.

***Inverse-Probability Weighting (IPW):*** The IPW estimator reconstructs samples from the complete cases by applying appropriate weights. Using Bayes' rule, the ground-truth

distribution is expressed as:

$$p(X_{(1)}, Y) = \frac{p(R = \vec{1}, X_{(1)}, Y)}{\pi(R = \vec{1} \mid X_{(1)}, Y)}.$$

This formulation is valid if the propensity score $\pi(R = \vec{1} \mid X_{(1)}, Y) = p(R = \vec{1} \mid X_{(1)}, Y)$ is identifiable, as $p(R = \vec{1}, X_{(1)}, Y) = p(R = \vec{1}, X, Y)$ is already a function of the observed data. When the missingness mechanism is MCAR, this estimator simplifies to complete case analysis.

***Multiple Imputation (MI):*** The MI estimator reconstructs $p(X_{(1)}, Y)$ by decomposing $X_{(1)}$ into its observed components $X_o$[1] and missing components $X_m$, as follows:

$$p(X_{(1)}, Y) = \sum_R p(X_m \mid X_o, Y, R) p(X_o, Y, R).$$

Here, $p(X_o, Y, R)$ corresponds to the observed data, while $p(X_m \mid X_o, Y, R)$ must be identified and estimated.

To ensure unbiased estimation, the imputation model must thus include the label $Y$ as an input. This was also verified in experiments (Moons et al., 2006). However, this introduces a practical concern: if the imputation model is misspecified, the imputed values may contain excessive information about $Y$, potentially leading to inflated prediction performance estimates.

To mitigate this risk, practitioners often exclude $Y$ from the imputation model (Srinivasan et al., 2016; Hapfelmeier, 2012; Ergul Aydin & Kamisli Ozturk, 2024). Imputing $X_m$ using $p(X_m \mid X_o, R)$ assumes $X_m \perp\!\!\!\perp Y \mid X_o, R$, implying that missing features contribute no additional information for predicting $Y$ beyond $X_o$ and $R$. Such a practice effectively dismisses the potential importance of missing features before the computation even begins, rendering the approach fundamentally flawed.

#### 3.4.2. Estimation of the Observed Data LOCO Metric

Estimating observed data feature importance metrics is more straightforward because the observed data LOCO metric is already identified. The primary requirement is ensuring that the classifier can appropriately handle missing values. Crucially, when assessing observed data feature importance metrics, no additional information should be introduced for the missing feature. This can be achieved either by applying the classifier directly (if it supports "?" values) or by using simple imputation methods (e.g., mean or median imputation) that do not inject additional predictive information. In the case of imputation, it is particularly important to include the unchanged missingness indicators $R$ in the classifier's input to preserve the informativeness of the missingness

---

[1]$X_o$ differs from $X$ because $X$ contains placeholders (e.g., "?" for missing values) that encode information about $R$.

pattern. While mean imputation is often considered a biased approach (Mera-Gaona et al., 2021), this is true only when a full data feature importance method is of interest. For observed data feature importance, mean imputation remains unbiased. A more rigorous mathematical justification for this result is provided in Appendix C.

Other popular approaches to estimate feature importance metrics can handle missing data intrinsically. These include for example tree-based methods such as XGBoost (Vo et al., 2024; Chen & Guestrin, 2016) and are often based on a principle known as missingness incorporated in attributes (MIA) (Twala et al., 2008). These methods merely handle the missing values within the classification and do not imagine counterfactual, "what if everything had been observed", scenarios. They are therefore also included in the observed data feature importance category.

### 3.4.3. Biased Estimation Methods

A common but highly problematic approach to computing feature importance metrics is conditional mean imputation. This method replaces missing values with

$$\mathbb{E}[X_m \mid X_o, Y, R],$$

or, more commonly (when $Y$ is excluded), with

$$\mathbb{E}[X_m \mid X_o, R].$$

While this simplification may appear reasonable, it introduces substantial bias when the goal is to estimate either full data or observed data feature importance metrics. We provide a detailed mathematical justification for a range of feature importance metrics in Appendix C and offer some intuition below.

Including $Y$ may create a deterministic relationship between $Y$ and the imputed values, potentially allowing a classifier to exploit this relationship and achieve artificially high prediction accuracy. Excluding $Y$, on the other hand, can still introduce a deterministic relationship between $X_m$ and $X_o$, thereby transferring predictive information from $X_o$ to the imputed values. This severely skews feature importance estimates. This method is thus neither suited to estimate full data, nor observed data feature importance metrics. Note that the same holds for all similar imputation methods that do not model the full density for the imputed features. This includes for example KNN-imputation (Zhang, 2012) which is also frequently used to report feature importance under missing data (Zhou et al., 2023).

In conclusion, when using imputation for the estimation of $LOCO^{FD}$ and $LOCO^{OD}$, the missing values must be imputed in a manner consistent with the classifiers respective "working conditions." For $LOCO^{FD}$, the classifier operates under full data availability. Therefore, missing values must be imputed using samples from the full data distribution. When

done correctly, $LOCO^{FD}$ is evaluated on a dataset that mirrors the original dataset as if no missingness had occurred. In this case, the classifier can leverage the imputed values to potentially improve performance. For $LOCO^{OD}$, evaluation occurs under conditions where missingness is present. Since the observed data already reflects the classifier's working conditions, no imputation is required. However, one may opt for impute-then-regress classifiers, where imputation serves purely as a means to handle missing values elegantly. In such cases, the imputation step does not introduce additional information beyond what is already present in the observed input features.

*Remark* 3.2. While the LOCO metrics in Eqs. 2 and 3 are valid for any classifier $f_{cl}$, the choice of training method is essential for interpretability. Classifiers must be trained for the specific conditions under which they will be deployed. To evaluate observed data feature importance metrics, the classifier must perform optimally in the presence of missing data. In contrast, full data feature importance metrics require a classifier trained for settings where all features are available. The implications of this distinction are discussed in Appendix D.

## 4. Feature Measurement Importance Gradient

The observed data and full data LOCO metrics assess the impact of completely omitting $X_j$ or $X_{(1),j}$ from the prediction model, providing an "all-or-nothing" perspective on feature importance. However, in practice, improving predictive performance often requires more fine-grained considerations. Rather than asking whether a feature is important in absolute terms, a more actionable question is: which feature, if measured more frequently, would yield the greatest improvement in prediction performance? For instance, if we could increase the measurement frequency of certain features, how should we prioritize them to maximize predictive gains? Answering this question provides practical guidance for optimizing data collection strategies.

To address this, we introduce the feature measurement importance gradient (FMIG), a novel metric that quantifies how prediction performance changes with small increases in the measurement probability of a given feature. We formalize FMIG by defining perturbation interventions on the measurement policy and discuss its identification and estimation.

Specifically, FMIG captures the sensitivity of predictive performance to marginal changes in measurement probability. To this end, we introduce a perturbation $\delta_j$ to the current measurement policy $\pi$, increasing the probability of measuring feature $j$. We denote the perturbed policy as $\pi_{\delta_j}$.

Given this perturbation, we can determine the gradient of the negative loss with respect to the perturbation to rank

features in their importance:

***Feature Measurement Importance Gradient (FMIG)*** $G_j$***:***

$$G_j = \nabla_{\delta_j} \mathbb{E}\left[-L\left(Y, f_{cl}(X_{(\pi_{\delta_j})}, R_{(\pi_{\delta_j})})\right)\right]\Big|_{\delta_j = \delta_j^*} \quad (4)$$

where $\delta_j^*$ is the value of $\delta_j$ that represents no perturbation. Here, $X_{(\pi_{\delta_j})}$ and $R_{(\pi_{\delta_j})}$ denote the counterfactual observed features and missingness indicators, had the perturbed policy $\pi_{\delta_j}$ been applied. We take the gradient with respect to the negative loss so that the FMIG quantifies the expected reduction in loss when the feature is measured slightly more frequently and thus leads to positive feature importances.

### 4.1. Measurement Process Assumptions

The definition of interventions on the measurement policy relies on the data's measurement process. Here, we consider a special MAR process. Features are measured sequentially, where the measurement of feature $j$ depends on all prior missingness indicators and observed features.

Under these assumptions, the measurement policy factorizes as:

$$\pi(R|X_{(1)}, Y) = \prod_{i=1}^{d} \pi_i(R_i|\underline{R}_{i-1}, \underline{X}_{i-1}), \quad (5)$$

where $\underline{R}_{i-1} \equiv \{R_1, \ldots, R_{i-1}\}$, $\underline{X}_{i-1} \equiv \{X_1, \ldots, X_{i-1}\}$ and we let $\underline{R}_0 \equiv \{\}$ and $\underline{X}_0 \equiv \{\}$.

### 4.2. Measurement Policy Interventions

To define perturbation interventions, we leverage the factorization from Eq. 5:

$$\pi_{\delta_j}(R|X_{(1)}, Y) =$$
$$= \pi_{j,\delta_j}(R_j|\underline{R}_{j-1}, \underline{X}_{i-1}) \prod_{i=1; i \neq j}^{d} \pi_i(R_i|\underline{R}_{i-1}, \underline{X}_{i-1}).$$

Here, the intervention modifies the measurement policy only for feature $j$. We adapt the perturbation from incremental propensity score interventions (Kennedy, 2019), defined as:

$$\pi_{j,\delta_j} = \frac{\delta_j \, \pi_j}{\delta_j \, \pi_j + 1 - \pi_j}, \quad \text{for } 0 < \delta_j < \infty,$$

where $\pi_j \equiv \pi_j(R_j = 1|\underline{R}_{j-1}, \underline{X}_{j-1})$. The parameter $\delta_j$ is the *odds ratio* of the intervention:

$$\delta_j = \frac{\pi_{j,\delta_j}/(1 - \pi_{j,\delta_j})}{\pi_j/(1 - \pi_j)} = \frac{\text{odds}_{\pi_{j,\delta_j}}(R_j = 1|\underline{R}_{j-1}, \underline{X}_{j-1})}{\text{odds}_{\pi_j}(R_j = 1|\underline{R}_{j-1}, \underline{X}_{j-1})}.$$

Values of $\delta_j < 1$ correspond to a decrease in the probability of measuring feature $j$, while $\delta_j > 1$ increases the probability. When $\delta_j = \delta_j^* = 1$, the original measurement policy is recovered: $\pi_{j,\delta_j=1} = \pi_j$.

### 4.3. Identification

Using the perturbation interventions, the feature measurement importance gradient $G_j$ can be identified as follows:

**Lemma 4.1.** *(Identification of the Feature Measurement Importance Gradient (FMIG))*
*Under SUTVA and the assumption about the missingness process from Eq. 5, the feature measurement importance gradient (FMIG) for feature $j$ is identified as:*

$$G_j = \nabla_{\delta_j} \mathbb{E}\left[-L\left(Y, f_{cl}(X_{(\pi_{\delta_j})}, R_{(\pi_{\delta_j})})\right)\right]\Big|_{\delta_j = 1}$$
$$= \mathbb{E}\left[-L\left(Y, f_{cl}(X, R)\right)(-1)^{1-R_j}(1 - \pi_j(R_j|\underline{R}_{j-1}, \underline{X}_{j-1}))\right].$$

The proof is provided in Appendix E. It demonstrates why, unlike full data feature importance metrics, the FMIG does not require a positivity assumption due to the chosen perturbation. This allows for far more reliable estimation in real-world settings with high fractions of missing values.

### 4.4. Estimation

The functional $G_j$, as identified in Lemma 4.1, is straightforward to estimate. It requires only the measurement policy of the feature of interest to be learned. An estimator is given by:

$$\hat{G}_j = \hat{\mathbb{E}}_n\left[-L\left(Y, f_{cl}(X, R)\right)(-1)^{1-R_j}(1 - \hat{\pi}_j(R_j|\underline{R}_{j-1}, \underline{X}_{j-1}))\right],$$

where $\hat{\mathbb{E}}_n$ is the empirical mean over a dataset of size $n$, and $\hat{\pi}_j$ is a model learned to approximate $\pi_j$. When using flexible machine learning models for $\hat{\pi}_j$, these should be trained on a separate data split. To restore full data efficiency, a cross-fitting scheme can be employed (Chernozhukov et al., 2018).

We extend the theory of the feature measurement importance gradient to time-series settings in Appendix F. In this setting, each feature is assumed to be measured at fixed time points, with predictions made for a time-series label at each time point.

## 5. Experiments

Missing data estimation methods cannot be tested on real-world data with real missingness due to the unavailability of ground truth features $X_{(1)}$. To address this limitation, we perform a series of synthetic experiments to illustrate the differences between feature importance metrics, the impact of positivity violations, and the significance of appropriate estimation methods.

### 5.1. Experiment Design

We consider a time-series prediction task where features $X^t \in (\mathbb{R} \cup \{"?"\})^d$ ($t \in \{1, 2, 3\}$) are used to predict labels

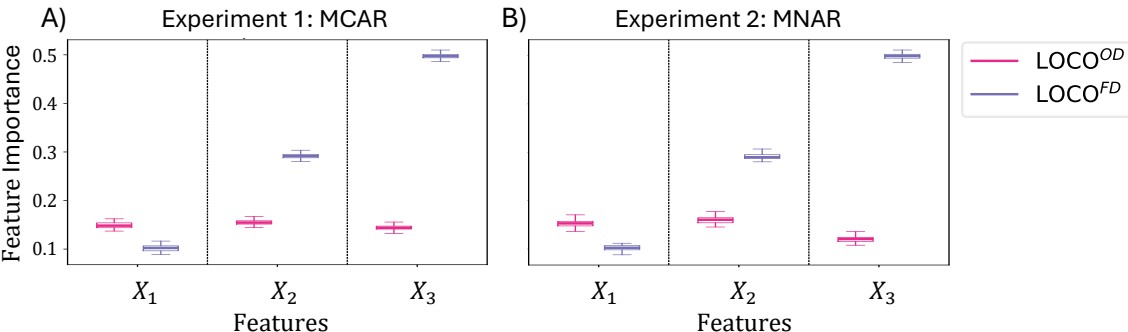

*Figure 1.* Comparison of feature importance metrics (ground truth estimates). A) Experiment 1: The full data LOCO metric increases from $X_1$ to $X_3$, reflecting the features' growing predictive importance. However, the observed data LOCO remains flat because the increasing predictive value is offset by higher missingness rates from $X_1$ to $X_3$. B) Experiment 2: Due to MNAR missingness ($X_{(1),3} \rightarrow R_1, R_2$), the observed data LOCO of $X_3$ is further reduced because the missingness mechanism induces correlations between $X_{(1),3}$ and $X_1$, $X_2$ via $R_1$, $R_2$. Confidence intervals were computed using the non-parametric bootstrap with 50 samples.

$Y^t$ at each time step. We use an "impute-then-regress" classifier with zero imputation and a temporal convolutional network (TCN) (Bai et al., 2018). Missingness mechanisms (MCAR, MNAR) are introduced at varying missingness rates across experiments. Full experimental details are provided in Appendix G. Additionally, Appendix H presents an experiment highlighting the importance of training classifiers appropriately for the metric under evaluation.

### 5.2. Synthetic Experiments and Results

**Experiment 1: Rarely measured variables have reduced observed data feature importance**

**Design:** We evaluate three variables with increasing predictive importance but decreasing measurement probabilities under an MCAR scenario: $p(R_1^t = 1) = 0.75$, $p(R_2^t = 1) = 0.5$, $p(R_3^t = 1) = 0.3$.

**Results:** While the full data LOCO reflects the increasing predictive importance from $X_1$ to $X_3$, the observed data LOCO remains flat (Figure 1A). This occurs because the gains in predictive performance from more informative features are counteracted by their higher missingness rates. The experiment illustrates how observed data feature importance metrics lead to different conclusions than full data feature importance metrics.

**Experiment 2: Informative missingness affects observed feature importance**

**Design:** Using the same dataset as Experiment 1, we simulate MNAR missingness where $X_{(1),3}^t$ influences missingness of features at subsequent time points. Average missingness probabilities remain equal to Experiment 1 with $p(R_1^t = 1) \approx 0.75$, $p(R_2^t = 1) \approx 0.5$, $p(R_3^t = 1) = 0.3$.

**Results:** Figure 1B) shows that the full data LOCO remains unchanged. However, the observed data LOCO of $X_{(1),3}$

decreases as its information is "leaked" into correlated variables ($R_1$, $R_2$). The experiment highlights how informative missingness redistributes feature importance in observed data.

### Experiment 3: Positivity violations introduce bias in estimation

**Design:** Using the same dataset as Experiment 1, we simulate stronger MCAR missingness: $p(R_1^t = 1) = 0.4$, $p(R_2^t = 1) = 0.2$, $p(R_3^t = 1) = 0.1$. This results in a near-complete absence of fully observed cases, effectively violating the positivity assumption.

**Results:** Figure 2 illustrates biases in estimating components of the full data LOCO metric. The complete case analysis, which is generally unbiased under MCAR, produces unbiased estimates in Figure 2A) for Experiment 1. However, in Experiment 3, where the positivity assumption is violated, the complete case estimator exhibits strong bias (Figure 2B). These results highlight the limitations of standard estimation methods for the full data LOCO metric when positivity is violated.

### Experiment 4: FMIG provides additional insights

**Design:** We evaluate three variables with equal full data feature importance, using an MCAR mechanism. We have $p(R_1^t = 1) = 1$, $p(R_2^t = 1) = 0.5$ and a constant feature $X_{(1),3}$ with $p(R_3^1 = 1) = 1$ and $p(R_3^t = 1) = 0.5$ for $t \geq 2$.

**Results:** Figure 3 illustrates the FMIG metric is zero for $X_1$ as it is already fully observed. FMIG is positive for $X_2$ which could, if measured more often, improve prediction performance. Measuring the constant $X_{(1),3}$ multiple times provides no predictive improvement. Since the additional feature measurement is completely uninformative, the FMIG even becomes slightly negative. The insight that measuring $X_{(1),3}$ more often will not improve prediction per-

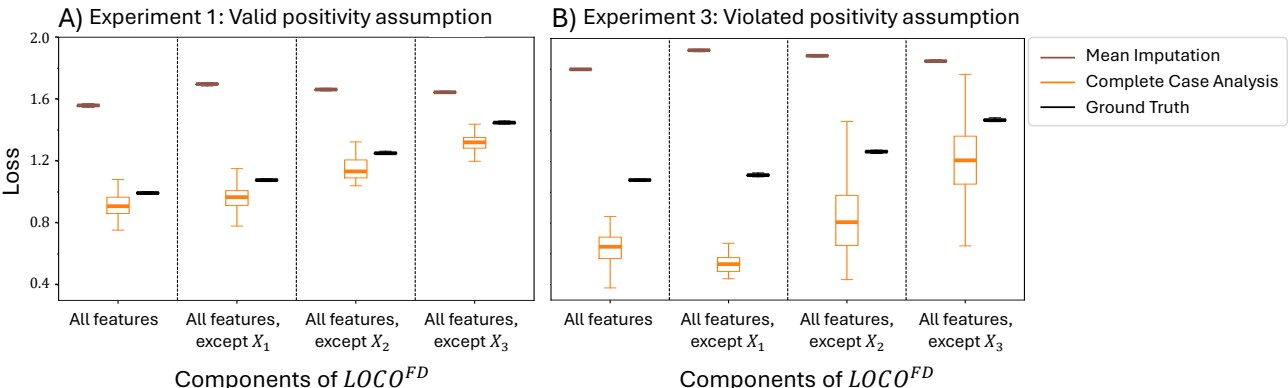

*Figure 2.* Effects of the positivity assumption violations on the estimation of full data LOCO components. The plot shows estimates using different estimators for $\mathbb{E}\left[-\sum_{t=1}^{T} L\left(Y^t, f_{cl}(\underline{X}_{(1)}^t, \underline{R}^t = \vec{1})\right)\right]$ (denoted as "All features") and $\mathbb{E}\left[-\sum_{t=1}^{T} L\left(Y^t, f_{cl,-j}(\underline{X}_{(1),-j}^t, \underline{R}_{-j}^t = \vec{1})\right)\right]$ for each feature (denoted as "All features, except $X_1$", "All features, except $X_2$", and "All features, except $X_3$"). A) Experiment 1: Under a valid positivity assumption, the complete case estimator provides an acceptable estimate of the ground truth. B) Experiment 3: Under violations of the positivity assumption, the complete cases analysis provides a biased estimate of the ground truth. Note that the classifiers used to evaluate feature importance metrics were trained on the observed data from each experiment. As a result, even the ground truth estimates differ slightly, with a lower loss in Experiment 1 due to less missingness during classifier training. Confidence intervals were computed using the non-parametric bootstrap with 50 samples.

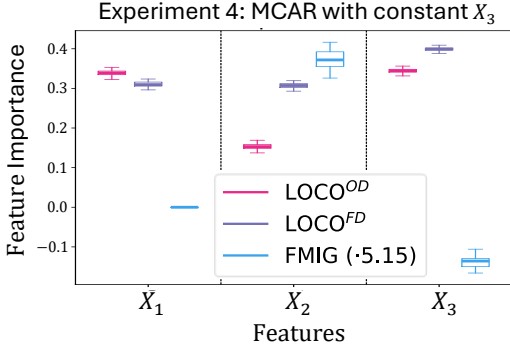

*Figure 3.* $X_{(1),3}$ is constant and fully observed at $t = 1$, but less frequently at later times. The feature measurement importance gradient FMIG is low for this feature, even though it has high observed data and full data feature importance.

formance is not apparent from observed or full data LOCO metrics, but only from the FMIG.

## 6. Discussion & Conclusion

In this work, we examined feature importance in the presence of missing data. Full data feature importance metrics, which assume no missingness, offer insights into the data-generating process but are often difficult to estimate and impractical when missingness persists at deployment. To address this, we introduced observed data feature importance metrics, which assess importance under a given measurement policy and highlighted key differences between these metrics, categorizing estimation methods based on

their target estimands.

Neither observed nor full data feature importance directly informs data collection. To address this, we introduce the feature measurement importance gradient (FMIG), a novel metric that identifies which features, if measured more frequently, would yield the greatest improvement in predictive performance. Synthetic experiments illustrate the distinct insights these metrics provide, emphasizing the need for tailored approaches in missing data settings. FMIG is thus not an alternative to the observed data or full data LOCO metrics, but an additional tool to assess the importance of a feature under a change in measurement probability.

While FMIG provides information on which features should be measured more frequently, it does not prescribe an optimal measurement policy. Active feature acquisition (AFA) (von Kleist et al., 2025; 2023), on the other hand, explicitly balances measurement costs against predictive value to derive optimal policies. FMIG thus complements AFA by identifying features worth prioritizing for optimization.

Our experimental evaluation is limited to synthetic data. Demonstrating the usefulness of feature importance metrics in real-world settings particularly where prior knowledge or separately collected, fully observed datasets are available is an important direction for future work. Such demonstrations could highlight the practical relevance of these metrics and inform their application in domains with varying measurement constraints.

## Acknowledgements

Joshua Wendland was supported by the European Research Council (ERC) Starting Grant DEUCE, grant no. 101077178. Ilya Shpitser acknowledges support from grants ONR N000142412701 and NSF CAREER 1942239. Carsten Marr acknowledges funding from the ERC under the European Unions Horizon 2020 research and innovation program (grant agreement no. 866411) and support from the Hightech Agenda Bayern.

## Impact Statement

Our method enhances the explainability of AI models by providing a principled way to assess feature importance under missing data, thereby improving trust and reliabilityespecially in high-stakes applications such as medical decision-making. By clarifying biases in existing methods, our work contributes to more transparent, interpretable, and fair AI systems, ultimately fostering more informed and responsible deployment of machine learning models.

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

## A. Background on Feature Importance Metrics

Feature importance metrics can be categorized into local and global approaches (Ewald et al., 2024). Local metrics focus on explaining individual predictions, while global metrics provide insights into the aggregate importance of features in the prediction process. This work centers on global, model-agnostic methods that can be applied to any predictive model.

Feature importance metrics assign an importance value to each feature, reflecting its contribution to prediction performance. Beyond their role in explaining model behavior, feature importance metrics are often used to derive reliable, population-level insights about the data-generating process. They are also closely linked to feature selection methods, as both seek to evaluate the relevance of features within a predictive model. In scenarios involving missing data, the challenges faced in assessing feature importance and selecting features often overlap, allowing similar methodologies to be applied across these tasks.

The field of feature importance metrics is dominated by three main approaches (Ewald et al., 2024). Permutation feature importance evaluates the impact of randomly permuting a features values on model performance. It is model-agnostic and widely used for its simplicity but may underestimate feature importance when features are highly correlated. Marginalization-based methods assess importance by marginalizing out the feature of interest, often requiring strong assumptions about the data-generating process to be valid. Finally, model refitting methods compare model performance before and after removing specific features. Techniques like LOCO (Lei et al., 2018) fall into this category, directly assessing the dependency of predictions on a given feature.

## B. The Leave-One-Covariate-Unmeasured (LOCU) Metric

In this appendix, we introduce an alternative estimand to the observed data LOCO metric, $LOCO^{OD}$. Unlike $LOCO^{OD}$, which evaluates the impact of leaving one covariate out of the input to the classifier, this new metric considers the scenario where the measurement of a covariate is entirely omitted.

We denote this alternative metric as *leave-one-covariate-unmeasured (LOCU)*. It is defined as follows:

*LOCU:*

$$LOCU = \mathbb{E}\left[L\left(Y, f_{cl}(X_{(-j)}, R_{(-j)})\right)\right] - \mathbb{E}\left[L\left(Y, f_{cl}(X, R)\right)\right],$$

where $X_{(-j)}$ denotes the counterfactual value $X$ would have taken if $R_j = 0$. This represents a scenario where all features are measured according to the current measurement policy, except for feature $j$, which is no longer measured. LOCU thus quantifies the change in prediction loss if feature $j$ is omitted from measurement. This can provide valuable insights for data collection decisions. For instance, in medical settings, if a particular test is costly, the metric can help assess the impact of discontinuing the test on prediction performance.

LOCU differs from the observed data LOCO metric $LOCO^{OD}$ in cases where the measurement process of other features depends on the measurement of feature $j$ (i.e., there are causal arrows from $R_j$ to other missingness indicators). For example, suppose feature $i$ is measured only if a test on feature $j$ yields a positive result. In this situation, omitting the measurement of feature $j$ would also prevent the measurement of feature $i$, potentially leading to a more substantial change in prediction performance.

While causal inference methods can be used to identify and estimate LOCU in such scenarios, this aspect is considered outside the scope of this work.

## C. Proofs for Unbiased Estimation Results for the LOCO Metrics

In this Appendix, we add mathematical proofs that complement the discussion about estimation for $LOCO^{OD}$ and $LOCO^{FD}$. We define $\theta$ as a general estimand and $\hat{\theta}$ as its corresponding estimator. The bias of the estimator is given by:

$$\text{Bias}(\hat{\theta}) = \mathbb{E}[\hat{\theta}] - \theta.$$

Our results extend to a broader class of full data feature importance metrics of the form:

$$\theta^{FD} = \mathbb{E}\left[g(\{f_{cl,s}(X_{(1),s}, R_s = \vec{1}) : s \in \mathscr{S}\}, Y)\right],$$

where $g$ can be any function and $\mathscr{S}$ represents the set of all feature subsets. This definition encompasses a wide range of metrics, including LOCO and Shapley values. Notably, Shapley values can be expressed as a weighted average of LOCO values across submodels (Verdinelli & Wasserman, 2023). Additionally, we consider observed data feature importance metrics $\theta^{OD}$ of the same form, with $X_{(1),s}$ and $R_s = \vec{1}$ replaced by $X_s$ and $R_s$.

## C.1. Conditional Mean Imputation for Full Data Feature Importance Metrics

Firstly, we demonstrate that conditional mean imputation results in biased estimation of full data feature importance metrics. Our analysis is based on the formulation used for the unbiased multiple imputation estimator:

$$
\begin{aligned}
\theta^{FD} &= \sum_{X_{(1)},Y} g(\{f_{cl,s}(X_{(1),s}, R_s = \vec{1}) : s \in \mathscr{S}\}, Y) p(X_{(1)}, Y) \\
&= \sum_{X_m, X_o, Y, R} g(\{f_{cl,s}(X_{m \wedge s}, X_{o \wedge s}, R_s = \vec{1}) : s \in \mathscr{S}\}, Y) p(X_m | X_o, Y, R) p(X_o, Y, R),
\end{aligned}
$$

where $p(X_m | X_o, Y, R)$ represents the imputation density. When the imputation model is learned, one can apply Monte Carlo integration to obtain an unbiased estimator $\hat{\theta}^{FD}$ for $\theta^{FD}$.

Conditional mean imputation, however, simplifies the above expression to:

$$
\theta^{FD} \approx \sum_{X_m, X_o, Y, R} g(\{f_{cl,s}(\mathbb{E}[X_{m \wedge s} | X_o, Y, R], X_{o \wedge s}, R_s = \vec{1}) : s \in \mathscr{S}\}, Y) p(X_o, Y, R),
$$

which assumes that the expectation operator can be pulled inside the functions $g$ and $f_{cl}$. This assumption is only valid if both functions are linear, which is generally not the case. Consequently, the use of conditional mean imputation introduces bias.

## C.2. Mean and Conditional Mean Imputation for Observed Data Feature Importance Metrics

Next, we demonstrate that mean imputation (or a broader class of imputation methods) does not introduce bias for observed data feature importance metrics $\theta^{OD}$. Since $\theta^{OD}$ is defined as a function of the classifier $f_{cl}$, the reported feature importance metric reflects classifier-specific importance rather than a general global measure.

A commonly used class of classifiers, referred to as impute-then-regress classifiers (Le Morvan et al., 2021), first impute the missing values and subsequently classify. If the classifier is sufficiently flexible and the dataset is large enough, the choice of imputation method becomes inconsequential, as no new information is introduced. Thus, any classifier that maps $X_s$ and $R_s$ to $Y$ can be used, including those employing mean imputation, without inducing bias.

However, this conclusion holds only if imputation is performed within the classifier itself using only its input features. If conditional mean imputation is applied to the entire dataset before choosing the subset for the classifier, bias arises. Let the imputed features be:

$$
X_i' = \begin{cases} X_i & \text{if } R_i = 1, \\ \hat{\mathbb{E}}_n[X_i | X_o, R = \vec{1}] & \text{if } R_i = 0. \end{cases}
$$

In this case, the imputed variable $X_i'$ effectively becomes a function of other features, $X_i' \equiv f(X_o, R = \vec{1})$. The resulting bias in the estimator is given by:

$$
\begin{aligned}
\text{Bias}(\hat{\theta}^{OD}) &= \mathbb{E}[\hat{\theta}^{OD}] - \theta^{OD} \\
&= \mathbb{E}\left[\hat{\mathbb{E}}_n[g(\{f_{cl,s}(X_s', R_s) : s \in \mathscr{S}\}, Y)]\right] - \mathbb{E}\left[g(\{f_{cl,s}(X_s, R_s) : s \in \mathscr{S}\}, Y)\right].
\end{aligned}
$$

This bias is generally nonzero because $X'_s$ depends on all features rather than just $X_s$. Consequently, applying conditional mean imputation before classification results in biased estimates.

## D. Impact of Classifier Training on LOCO Metrics

The LOCO metrics in Eqs. 2 and 3 highlight the importance of aligning classifier training with the conditions of deployment. Specifically:

Full data feature importance metrics require a classifier $f_{cl}^{FD}$ designed to handle fully available features:

$$f_{cl}^{FD}(X_{(1)}, R = \vec{1}) = p(Y \mid X_{(1)}, R = \vec{1}).$$

Observed data feature importance metrics require a classifier $f_{cl}^{OD}$ capable of handling missing data:

$$f_{cl}^{OD}(X, R) = p(Y \mid X, R).$$

Training these classifiers introduces unique challenges. For $f_{cl}^{FD}$, the absence of $X_{(1)}$ in the training data creates a missing data problem, complicating the training process. In contrast, training $f_{cl}^{OD}$ involves dealing with missing values ("?") in $X$. While some models inherently handle missing data (Twala et al., 2008), a common alternative is to impute missing values. However, this approach can introduce discontinuities, which hinder the learning process.

To address these challenges, "impute-then-regress" classifiers (Le Morvan et al., 2021) provide an effective solution. These classifiers leverage the following decomposition:

$$f_{cl}^{OD}(X, R) = p(Y \mid X_m = E[X_m \mid X_o, R], X_o, R).$$

Here, missing values ($X_m$) are replaced with their conditional expectations based on observed values $X_o$ (and missingness indicators $R$). This imputation is intrinsic to the classifier and differs from post hoc imputation used during feature importance evaluation. Conditional mean imputation is often a practical and effective choice in this context.

The improper training of classifiers can introduce additional bias in LOCO metric estimations, particularly for full data feature importance metrics when $f_{cl}^{OD}$ is used instead of $f_{cl}^{FD}$. These biases are exacerbated in missing data scenarios such as MAR and MNAR, compared to MCAR. Awareness of these limitations is crucial for ensuring robust and interpretable feature importance assessments.

## E. Proof of Lemma 4.1

In this appendix, we prove Lemma 4.1 which states the form of the feature measurement importance gradient.

*Proof.* The gradient simplifies as follows:

$$
\begin{aligned}
G_j &= \nabla_{\delta_j} \mathbb{E}\left[ -L\left(Y, f_{cl}(X_{(\pi_{\delta_j})}, R_{(\pi_{\delta_j})})\right) \right]\Big|_{\delta_j=1} = \\
&\overset{*_1}{=} \nabla_{\delta_j} \sum_{R,Y,X_{(1)}} -L\left(Y, f_{cl}(X,R)\right) \pi_{\delta_j}(R|X_{(1)}, Y) p(X_{(1)}, Y)\Big|_{\delta_j=1} = \\
&= \sum_{R,Y,X_{(1)}} -L\left(Y, f_{cl}(X,R)\right) \prod_{i=1}^{d} \pi_i(R_i|\underline{R}_{i-1}, \underline{X}_{i-1}) \nabla_{\delta_j} \pi_{j,\delta_j}(R_j|\underline{R}_{j-1}, \underline{X}_{j-1})\Big|_{\delta_j=1} p(X_{(1)}, Y) = \\
&\overset{*_2}{=} \sum_{R,Y,X_{(1)}} -L\left(Y, f_{cl}(X,R)\right) \prod_{i=1}^{d} \pi_i(R_i|\underline{R}_{i-1}, \underline{X}_{i-1}) \pi_j(R_j|\underline{R}_{j-1}, \underline{X}_{j-1})(-1)^{1-R_j}(1 - \pi_j(R_j|\underline{R}_{j-1}, \underline{X}_{j-1})) p(X_{(1)}, Y) = \\
&= \mathbb{E}\left[ -L\left(Y, f_{cl}(X,R)\right) \cdot (-1)^{1-R_j}(1 - \pi_j(R_j|\underline{R}_{j-1}, \underline{X}_{j-1})) \right].
\end{aligned}
$$

with the following explanations:

In ∗1) we use the g-formula (Robins, 1986), which requires the SUTVA assumption, exchangeability (ensured by the MAR assumption) and in general positivity. The last assumption of positivity is, however, always guaranteed due to the form of our intervention. Positivity in general necessitates that $\forall r_j, \underline{r}_{j-1}, \underline{x}_{j-1}$ with $p(\underline{R}_{j-1} = \underline{r}_{j-1}, \underline{X}_{j-1} = \underline{x}_{j-1}) > 0$ and $\pi_{j,\delta_j}(R_j = r_j | \underline{R}_{j-1} = \underline{r}_{j-1}, \underline{X}_{j-1} = \underline{x}_{j-1}) > 0$, that we must also have $\pi_j(R_j = r_j | \underline{R}_{j-1} = \underline{r}_{j-1}, \underline{X}_{j-1} = \underline{x}_{j-1}) > 0$. This always holds, as setting with $\pi_j = 0$ yield

$$\pi_{j,\delta_j} = \frac{\delta_j \cdot 0}{\delta_j \cdot 0 + 1 - 0} = 0,$$

settings with $\pi_j = 1$ give

$$\pi_{j,\delta_j} = \frac{\delta_j \cdot 1}{\delta_j \cdot 1 + 1 - 1} = 1.$$

and settings with $\pi_j \in (0,1)$ imply

$$\pi_{j,\delta_j} = \frac{\delta_j \cdot \pi_j}{\delta_j \cdot \pi_j + 1 - \pi_j} > 0$$

since $\delta_j \cdot \pi_j > 0$, and $\delta_j \cdot \pi_j + 1 - \pi_j > 0$ holds.

In ∗1), we also use sums for the integrals to keep notation simple, but for continuous $X$ one can replace the sums with proper integrals.

In ∗2), we used the following simplifications of the gradient of the perturbed measurement policy. We first look at the case $R_j = 1$:

$$\nabla_{\delta_j} \pi_{j,\delta_j}(R_j = 1 | \underline{R}_{j-1}, \underline{X}_{j-1})\Big|_{\delta_j=1} = \nabla_{\delta_j} \frac{\delta_j \cdot \pi_j}{\delta_j \cdot \pi_j + 1 - \pi_j}\Big|_{\delta_j=1} \overset{*2.1}{=} \frac{\pi_j(\delta_j \cdot \pi_j + 1 - \pi_j) - (\delta_j \cdot \pi_j) \cdot (\pi_j)}{(\delta_j \cdot \pi_j + 1 - \pi_j)^2}\Big|_{\delta_j=1} =$$

$$= \frac{\pi_j(1 - \pi_j)}{(\delta_j \cdot \pi_j + 1 - \pi_j)^2}\Big|_{\delta_j=1} = \pi_j(1 - \pi_j)$$

where ∗2.1) follows from the quotion rule of differentiation.

For $R_j = 0$, we find:

$$\nabla_{\delta_j} \pi_{j,\delta_j}(R_j = 0 | \underline{R}_{j-1}, \underline{X}_{j-1})\Big|_{\delta_j=1} = \nabla_{\delta_j}(1 - \pi_{j,\delta_j}(R_j = 1 | \underline{R}_{j-1}, \underline{X}_{j-1}))\Big|_{\delta_j=1} = -\nabla_{\delta_j} \pi_{j,\delta_j}(R_j = 1 | \underline{R}_{j-1}, \underline{X}_{j-1})\Big|_{\delta_j=1} =$$

$$= -\pi_j(1 - \pi_j) = -\Big(1 - \pi_j(R_j = 0 | \underline{R}_{j-1}, \underline{X}_{j-1})\Big)\pi_j(R_j = 0 | \underline{R}_{j-1}, \underline{X}_{j-1})$$

Bringing both together, we find the transformation in ∗2) and thus completes the proof.

$\square$

# F. Feature Measurement Importance Gradient for Time-series Settings

In this appendix, we extend the feature measurement importance gradient to time-series settings. For time-step $t$, we denote the observed features as $X^t \in (\mathbb{R} \cup \{"?"\})^d$ and all features up to time $t$ as $\underline{X}^t = \{X^1, \ldots, X^t\}$. The time-dependent label is $Y^t \in \{0,1\}$. A classifier predicts $Y^t$ at each time step. This classifier is $f_{cl}^t(\underline{X}^t, \underline{R}^t)$, where $\underline{R}^t$ denotes the missingness indicators up to $t$.

We redefine the FMIG for the time-series settings as:

*Feature Measurement Importance Gradient (FMIG) $G_j$:*

$$G_j = \nabla_{\delta_j} \mathbb{E}\left[\sum_{t=1}^{T} -L\left(Y^t, f_{cl}^t(\underline{X}_{(\pi_{\delta_j})}^t, \underline{R}_{(\pi_{\delta_j})}^t)\right)\right]\Bigg|_{\delta_j=\delta_j^*} \tag{6}$$

where $\delta_j^*$ is the value of $\delta_j$ that represents no perturbation.

## F.1. Measurement Process Assumptions

We make the following assumptions about the factorization of the missingness process:

$$\pi(R|X_{(1)},Y) = \prod_{t=1}^{T} \pi^t(R^t|\underline{R}^{t-1},\underline{X}^{t-1}), \tag{7}$$

where we assume the multivariate missingness indicators further factorize into conditionally independent terms:

$$\pi^t(R^t|\underline{R}^{t-1},\underline{X}^{t-1}) = \prod_{i=1}^{d} \pi_i^t(R_i^t|\underline{R}^{t-1},\underline{X}^{t-1}). \tag{8}$$

We thus assume that at each time-point, the measurement of feature $i$ will depend on all observed features in the past.

## F.2. Measurement Policy Interventions

To redefine perturbation interventions for the time-series setting by now jointly intervening on a feature at all time-steps. We obtain:

$$\pi_{\delta_j}(R|X_{(1)},Y) = \prod_{t=1}^{T} \pi_{j,\delta_j}^t(R_j^t|\underline{R}^{t-1},\underline{X}^{t-1}) \prod_{i\neq j}^{d} \pi_i^t(R_i^t|\underline{R}^{t-1},\underline{X}^{t-1}). \tag{9}$$

where each individual intervention follows the same odds ratio parameterization as in the main body.

## F.3. Identification

Using the perturbation interventions, the feature measurement importance gradient $G_j$ can be identified for the time-series setting as the following corollary states:

**Corollary F.1.** *(Identification of the Feature Measurement Importance Gradient (FMIG) for Time-series Settings)*
*Under SUTVA and the assumptions about the missingness process from Eqs. 7 and 8, the feature measurement importance gradient (FMIG) for feature $j$ is identified as:*

$$G_j = \nabla_{\delta_j} \mathbb{E}\left[\sum_{t=1}^{T} -L\left(Y^t, f_{cl}^t(\underline{X}_{(\pi_{\delta_j})}^t, \underline{R}_{(\pi_{\delta_j})}^t)\right)\right]\Bigg|_{\delta_j=1}$$

$$= \mathbb{E}\left[\sum_{t=1}^{T} -L\left(Y^t, f_{cl}^t(\underline{X}^t,\underline{R}^t)\right) \cdot \left(\sum_{k=1}^{t}(-1)^{1-R_j^k}(1-\pi_j^k(R_j^k|\underline{R}^{k-1},\underline{X}^{k-1}))\right)\right] \tag{10}$$

*Proof.* The gradient simplifies as follows:

$$
G_j = \nabla_{\delta_j} \mathbb{E}\left[\sum_{t=1}^{T} -L\left(Y^t, f_{cl}^t(\underline{X}^t_{(\pi_{\delta_j})}, \underline{R}^t_{(\pi_{\delta_j})})\right)\right]\Bigg|_{\delta_j=1} =
$$

$$
= \nabla_{\delta_j} \sum_{R,Y,X_{(1)}} \sum_{t=1}^{T} -L\left(Y^t, f_{cl}^t(\underline{X}^t_{(\pi_{\delta_j})}, \underline{R}^t_{(\pi_{\delta_j})})\right) \pi_{\delta_j}(R|X_{(1)}, Y) p(X_{(1)}, Y)\Bigg|_{\delta_j=1} =
$$

$$
\overset{*1}{=} \sum_{R,Y,X_{(1)}} \sum_{t=1}^{T} -L\left(Y^t, f_{cl}^t(\underline{X}^t, \underline{R}^t)\right) \left(\nabla_{\delta_j} \prod_{\tau=1}^{t} \pi_{j,\delta_j}^{\tau}(R_j^{\tau}|\underline{R}^{\tau-1}, \underline{X}^{\tau-1})\Bigg|_{\delta_j=1}\right) \prod_{\tau=1}^{t}\prod_{i=1}^{d} \pi_i^{\tau}(R_i^{\tau}|\underline{R}^{\tau-1}, \underline{X}^{\tau-1}) p(X_{(1)}, Y) =
$$

$$
\overset{*2}{=} \sum_{R,Y,X_{(1)}} \sum_{t=1}^{T} -L\left(Y^t, f_{cl}^t(\underline{X}^t, \underline{R}^t)\right) \sum_{k=1}^{t}\left(\prod_{\tau=1,\tau\neq k}^{t} \pi_{j,\delta_j}^{\tau}(R_j^{\tau}|\underline{R}^{\tau-1}, \underline{X}^{\tau-1}) \nabla_{\delta_j} \pi_{j,\delta_j}^{k}(R_j^{k}|\underline{R}^{k-1}, \underline{X}^{\tau-1})\Bigg|_{\delta_j=1}\right)
$$

$$
\cdot \prod_{\tau=1}^{t}\prod_{i=1}^{d} \pi_i^{\tau}(R_i^{\tau}|\underline{R}^{\tau-1}, \underline{X}^{\tau-1}) p(X_{(1)}, Y) =
$$

$$
\overset{*3}{=} \sum_{R,Y,X_{(1)}} \sum_{t=1}^{T} -L\left(Y^t, f_{cl}^t(\underline{X}^t, \underline{R}^t)\right) \sum_{k=1}^{t}\left(\prod_{\tau=1}^{t} \pi_j^{\tau}(R_j^{\tau}|\underline{R}^{\tau-1}, \underline{X}^{\tau-1})(-1)^{1-R_j^k}\left(1 - \pi_j^k(R_j^k|\underline{R}^{k-1}, \underline{X}^{k-1})\right)\right)
$$

$$
\cdot \prod_{\tau=1}^{t}\prod_{i=1}^{d} \pi_i^{\tau}(R_i^{\tau}|\underline{R}^{\tau-1}, \underline{X}^{\tau-1}) p(X_{(1)}, Y) =
$$

$$
= \sum_{R,Y,X_{(1)}} \sum_{t=1}^{T} -L\left(Y^t, f_{cl}^t(\underline{X}^t, \underline{R}^t)\right) \sum_{k=1}^{t}\left((-1)^{1-R_j^k}\left(1 - \pi_j^k(R_j^k|\underline{R}^{k-1}, \underline{X}^{k-1})\right)\right)
$$

$$
\cdot \prod_{\tau=1}^{t} \pi_j^{\tau}(R_j^{\tau}|\underline{R}^{\tau-1}, \underline{X}^{\tau-1}) \prod_{i=1}^{d} \pi_i^{\tau}(R_i^{\tau}|\underline{R}^{\tau-1}, \underline{X}^{\tau-1}) p(X_{(1)}, Y) =
$$

$$
= \mathbb{E}\left[\sum_{t=1}^{T} -L\left(Y^t, f_{cl}^t(\underline{X}^t, \underline{R}^t)\right) \cdot \left(\sum_{k=1}^{t}(-1)^{1-R_j^k}(1 - \pi_j^k(R_j^k|\underline{R}^{k-1}, \underline{X}^{k-1}))\right)\right].
$$

with the following explanations:

In $*1$), we used the fact that only the part of $\pi$ that models missingness for feature $j$ depends on the gradient. We further use the fact that $\sum_{t=1}^{T} -L\left(Y^t, f_{cl}^t(\underline{X}^t, \underline{R}^t)\right)$ is independent of any missingness indicators $R^{\tau}$ with $\tau > t$.

In $*2$), we apply the product rule of differentiation.

In $*3$), we leverage the same simplification of $\nabla_{\delta_j} \pi_{j,\delta_j}^{k}(R_j^k|\underline{R}^{k-1}, \underline{X}^{\tau-1})$ as was used in Appendix E for the simpler static feature setting. $\qquad\square$

# G. Experiment Details

In this appendix, we provide a detailed description of the experimental setup, including the data generation process, missingness mechanisms, training configurations, and the methods used to estimate feature importance metrics. These details align with the experiments presented in Section 5 and Figures 1, 2, 3 and 4.

## G.1. Experiment Setup

The experiments are based on a synthetic time-series dataset with three features ($d = 3$) and three time-points ($T = 3$). For time-step $t$, we denote the observed features as $X^t \in (\mathbb{R} \cup \{\text{"?"}\})^d$ and all features up to time $t$ as $\underline{X}^t = \{X^1, \dots, X^t\}$. The time-dependent label is $Y^t \in \{0, 1\}$. A classifier predicts $Y^t$ at each time step, and for the observed data, this classifier is $f_{cl}^{OD}(\underline{X}^t, \underline{R}^t)$, where $\underline{R}^t$ denotes the missingness indicators up to $t$.

The data generation process is defined as follows. Features $X^t_{(1),i}$ are independent across dimensions and generated recursively:

$$X^t_{(1),i} = \begin{cases} \gamma_i X^{t-1}_{(1),i} + (1-\gamma_i)\epsilon^t_i, & \text{if } t > 1, \\ \epsilon^t_i, & \text{if } t = 1, \end{cases}$$

where $\epsilon^t_i \sim \mathcal{N}(0, \sigma = 1)$ and $\gamma_i$ controls the temporal dependence for feature $i$. We let $X^0_{(1)} \equiv \vec{0}$. Labels $Y^t$ are binary and determined as:

$$p(Y^t = 1) = \begin{cases} 1, & \text{if } \sum_i W_i X^t_{(1),i} + \sum_i W_i X^{t-1}_{(1),i} > 0, \\ 0.2, & \text{otherwise.} \end{cases}$$

This label distribution simulates a scenario where not all data points are equally easy to classify. Parameters $W$ represent feature importances, which vary across experiments. We generate 100,000 data points and split them into 30% for training the classifier, 30% for training the measurement policy, and 40% for testing.

### G.2. Missingness Mechanisms

The missingness process in these experiments follows an MCAR or MNAR scenario. For the MNAR case, the missingness probabilities are modeled as:

$$\pi(R^t_i = 1 | X^{t-1}_{(1)}) = \sigma(\alpha_1 + \alpha_2 X^{t-1}_{(1)}),$$

where $\sigma$ denotes the logistic function. In this example, $X^{t-1}_{(1)}$ directly influences the missingness of $R^t_i$, making the process MNAR. The exact missingness probabilities used in each experiment are summarized in Table 1.

### G.3. Training and Evaluation

We used an "impute-then-regress" classifier (Le Morvan et al., 2021) with zero imputation and a temporal convolutional network (TCN) (Bai et al., 2018) to classify labels $Y^t$. The classifier uses four layers, with 32 channels per layer, a batch size of 2,000, dropout rate of 0.2, and a learning rate of 0.001.

Feature importance is evaluated using the observed and full data LOCO and the feature measurement importance gradient (FMIG). When we report the full data LOCO metric and do not mention the estimation method, it refers to the ground truth feature importance, as computed from the complete dataset without missingness. Binary cross-entropy is used as the loss function when calculating the LOCO metrics and the FMIG. We report confidence intervals using the nonparametric bootstrap with 50 samples. However, due to computational cost, we do not retrain the classifier for each bootstrap iteration. As a result, the confidence intervals may be overly optimistic.

### G.4. Experiment Configurations

The detailed configurations for each experiment, including the data-generating process parameters $(W, \gamma)$ and missingness mechanisms $(\pi)$, are provided in Table 1.

## H. Experiment Regarding Classifier Training Procedures

To illustrate the impact of proper classifier training, we modified the design of Experiment 3 by training the classifier on both the observed dataset (denoted as $f^{OD}_{cl}$) and the full dataset (denoted as $f^{FD}_{cl}$). This experiment, conducted under a violation of the positivity assumption and a high percentage of missingness, demonstrates how improper classifier training affects feature importance metrics. The results, shown in Figure 4, reveal that classifiers trained on observed data yield lower full-data LOCO metrics compared to those trained on the full dataset. Importantly, both LOCO metrics were evaluated on the ground truth dataset without missingness, highlighting how improper training introduces additional bias on top of the bias already inherent in estimating full data LOCO metrics from observed data. The consequences are expected to be even stronger for MAR and MNAR missingness scenarios, compared to this MCAR experiment.

These findings underscore the critical importance of aligning classifier training with the target metric to ensure accurate and meaningful feature importance evaluations. In practice, however, full alignment is often unfeasible due to the absence of ground truth data. Nonetheless, being aware of this limitation is crucial, as it allows for adjustments of training approaches to mitigate its consequences.

| Experiment | Data-Generating Process | Missingness Mechanisms |
|------------|------------------------|------------------------|
| Exp 1 | $W_1 = \frac{1}{6}; \gamma_1 = 0.2$ | $\pi(R_1^t = 1) = 0.75$ |
|  | $W_2 = \frac{2}{6}; \gamma_2 = 0.2$ | $\pi(R_2^t = 1) = 0.5$ |
|  | $W_3 = \frac{3}{6}; \gamma_3 = 0.2$ | $\pi(R_3^t = 1) = 0.3$ |
| Exp 2 | $W_1 = \frac{1}{6}; \gamma_1 = 0.2$ | $\pi(R_1^t = 1 \mid X_{(1),2}^{t-1}) = \sigma(1.7 - 3.0 X_{(1),2}^{t-1})$ |
|  | $W_2 = \frac{2}{6}; \gamma_2 = 0.2$ | $\pi(R_2^t = 1 \mid X_{(1),2}^{t-1}) = \sigma(-3.0 X_{(1),2}^{t-1})$ |
|  | $W_3 = \frac{3}{6}; \gamma_3 = 0.2$ | $\pi(R_3^t = 1) = 0.3$ |
| Exp 3 | $W_1 = \frac{1}{6}; \gamma_1 = 0.2$ | $\pi(R_1^t = 1) = 0.4$ |
|  | $W_2 = \frac{2}{6}; \gamma_2 = 0.2$ | $\pi(R_2^t = 1) = 0.2$ |
|  | $W_3 = \frac{3}{6}; \gamma_3 = 0.2$ | $\pi(R_3^t = 1) = 0.1$ |
| Exp 4 | $W_1 = \frac{1}{3}; \gamma_1 = 0.2$ | $\pi(R_1^t = 1) = 1.0$ |
|  | $W_2 = \frac{1}{3}; \gamma_2 = 0.2$ | $\pi(R_2^t = 1) = 0.5$ |
|  | $W_3 = \frac{1}{3}; \gamma_3 = 1.0$ | $\pi(R_3^1 = 1) = 1.0; \pi(R_3^t = 1) = 0.5$ (for $t > 1$) |

*Table 1.* Experiment-specific details of the data-generating processes and missingness mechanisms.

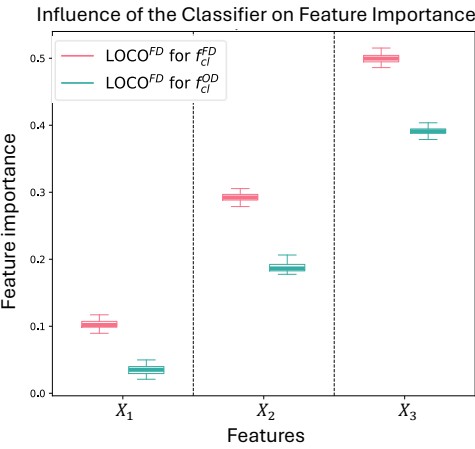

*Figure 4.* Impact of classifier training on feature importance: The full data LOCO metric derived from a classifier trained on observed data underestimates feature importance compared to a classifier trained on ground-truth data without missingness. Both LOCO metrics are estimated on the ground truth data.

