# OpenReview forum: "Feature Importance Metrics in the Presence of Missing Data"
_ICML.cc/2025/Conference — ICML 2025 poster_

### Official Review · Reviewer_3Vm4 · 2025-03-13

**Overall Recommendation:** 4

**Summary:**

This paper tackles the challenge of determining feature importance in realistic scenarios where data is missing. It is the first to explicitly formulate this problem and in doing so, introduces FMIG, a novel gradient-based metric that quantifies how small increases in the frequency of feature measurement can improve prediction performance. The approach is backed by theoretical derivations and validated through synthetic experiments under various missing data scenarios.

## update after rebuttal
After reading the other reviewers' comments and the authors' rebuttal, I have decided to keep my original score. I appreciate the authors’ thoughtful response to my review. While I still believe it may be possible to obtain or semi-generate a more realistic setting to evaluate the proposed method, I think the paper meets the current experimental standards for more theoretical works at this conference and should be accepted.

**Claims And Evidence:**

The paper clearly lays out its theoretical derivations and presents synthetic experiments that support its main claims. It also clearly defines the theoretical settings in which the method applies (MAR vs. NMAR). However, the claim that FMIG can effectively guide data acquisition in practice is less supported, since all experiments are synthetic. The assumption that the gradient approach can accuractly capture the effects on data in the real world need to be validated.

**Essential References Not Discussed:**

I'm not familiar enough with the latest works in that field to provide a solid review or opinion on this part.

**Experimental Designs Or Analyses:**

As I mentioned in the Methods and Evaluation section, the synthetic experiments are well demonstrating the differences between full and observed data scenarios, as well as the insights that could be captured by FMIG. However, further testing on real-world data would help validate these findings in practice.

**Methods And Evaluation Criteria:**

The methods and evaluation criteria are reasonable for an initial validation of the proposed approach. The paper’s synthetic experiments help illustrate how both the full data and observed data metrics, as well as the FMIG, behave under controlled conditions. However, further testing on benchmark or real-world datasets would be necessary to confirm the practical applicability of these methods.

One useful suggestion would be to induce synthetic missingness in real-world data by syntactically perturbing the missingness mechanism. Additionally, it would be beneficial to run an experiment with real dataset where features are missing and where there is prior knowledge of the expected effects. This would allow the authors to assess whether the insights provided by FMIG and the other metrics align with known expectations, even without direct access to the full data or comparison to the full data scenario.

**Other Comments Or Suggestions:**

-

**Other Strengths And Weaknesses:**

In addition to the strengths mentioned above, I would like to emphasize one more strength: the use of the MAR formulation is well justified  as it provides a robust model for real-world scenarios where sampling decisions are often based on the outcomes of other features. Ideally having a method for NMAR could be very useful, but as stated much harder to tackle.

**Questions For Authors:**

-

**Relation To Broader Scientific Literature:**

In my view, this paper makes a very important contribution, as the entire field of feature importance has generally focused only on the case where all data is always available which is very often not true in practice. The examples provided, especially in medical contexts, are highly relevant and highlight the need for further research in this area. Moreover, the authors offer an initial solution backed by a strong theoretical foundation, which I believe could pave the way for both theoretical and empirical research with significant impact.

**Theoretical Claims:**

I have briefly checked the main claims and they seem valid. In particular, the proofs related to the identification of FMIG and the derivation of the LOCO metrics appear solid. The theoretical ground is strong, and the settings and assumptions are clearly stated and highlighted as expected.

---

> ### Author Rebuttal · Authors · 2025-03-31
>
> We greatly appreciate the reviewer's thoughtful review and positive feedback. We recognize the value of conducting additional experiments, particularly those utilizing datasets with established prior knowledge of expected effects, to further validate our results. Unfortunately, we do not have access to such datasets. As a result, we will explicitly address this limitation in the Conclusion section, highlighting the potential for future research to explore these opportunities.

---

### Official Review · Reviewer_sya3 · 2025-03-13

**Overall Recommendation:** 3

**Summary:**

## update after rebuttal: 2 --> 3


When applying feature importance (FI) methods as explanation techniques for machine learning (ML) models, the presence of missing data is typically not considered. The authors highlight the issue that missing values can impact FI method results and demonstrate this using the leave-one-covariate-out (LOCO) approach. They adapt LOCO to account for missing values by incorporating the missingness process into the computation. Furthermore, the authors introduce the new "feature measurement importance gradient" (FMIG) as a novel metric to assess how the increased measurement or observation of a feature would influence predictive performance. They support their claims with several synthetic examples.

**Claims And Evidence:**

While I agree that the issue the authors raise here is an interesting one, especially for applied data science work, the paper is often not well written (especially but not limited to formal notation, see further below) and the experimental results are not extremely insightful.

**Essential References Not Discussed:**

See above.

**Experimental Designs Or Analyses:**

General comment w.r.t. exp setup: I am not exactly sure why the authors mix in time-dependent components in their experiment without discussing them before in the paper (with only a short ref to the appendix where this is covered more). This is certainly not "illegal" but somewhat unmotivated and / or one wonders why the i.i.d. scenario is not explored at all. Also, the experiments are not very "large" (I do not mean data size here!), so not very many different scenarios are explored, so it is unclear how much the results generalize.

Experiments 1 and 2: The experiment setup seems ok to me, but the experiments pretty much say what we knew already? Missingness reduces the predictive value of features. And "informed missingness", well…. "informs", so it influences our loss-based LOCO result.

Experiment 3: I would rather call this more or less a "sanity check" for the FMIG measure. Which is fine. That LOCO cannot provide the same information seems clear, as it was never constructed for this. LOCO assesses a feature's importance, FMIG reflects the importance or added value of increasing a feature’s measurement probability.

Experiment 4: The presentation is somewhat confusing, as it initially follows the scenario from Experiment 1 but is later adapted in the results section. Additionally, the ground truth is unclear (as it differs between Figures 2A and 2B).

Detailed information about the experimental setup is given in the appendix, but primarily for experiments 1+2; experiments 3+4 lack detailed information.

**Methods And Evaluation Criteria:**

LOCO is examined from three different perspectives: (1) full data LOCO, (2) observed data LOCO, and (3) leave-one-covariate-unmeasured (LOCU), with a primary focus on the first two.

The distinction between these perspectives is not entirely clear, leaving some open questions. For instance, both LOCO^FD and LOCO^OD can be approximated by imputing missing values and differ only in the imputation method used. This raises the question of how to distinguish between simple and non-simple imputation methods.

Furthermore, concrete application examples illustrating when each perspective is of particular interest would be valuable, especially in differentiating between LOCO^OD and LOCU.

**Other Comments Or Suggestions:**

Appendix F, line 938: "eps_i" is used twice here, in the initialization line (2nd) and the recursion (1st). I would GUESS these should be independent noise variables? So different? Defined like this, we convex-combine the same stuff and the recursion would never change the value of X^t ??

According to Table 1, Appendix F.4, the probability of observing X3 in experiment 1 is 0.25; in the experiment description in section 5.2 for experiment 1 it is said to be 0.3. Which number is correct?

Further suggestions for improvements:
A. The role of imputation in missing data should be mentioned in the introduction.
B. The approach is only formulated for classification. Why? I see no reason for that restriction and the authors do not even discuss it.
C. I would highly appreciate a simple example clarifying FMIG.

**Other Strengths And Weaknesses:**

The paper has severe problems in notation and description. This makes it sometimes very hard to follow, and the burden on the reader too heavy. One should not have to "guess" things means. Some examples:

Cross-entropy is not defined on "labels" x "labels", but "label" x "probs". For the same reason, the classifier f_cl also cannot output labels but must output probs.

It is often confusing that the authors conflate the "training procedure" of the model with the "predictor" of the model. I know that it can be annoying to distinguish between the two in notation, but it would help.
Just look at formula (2): The authors write f(X_{(1), -j}). First of all, by definition, f would always take R. Here, it does not. Then, via "-j" the input loses one dimension. The definition of f does not allow this. I can GUESS what is LIKELY meant here —> But I should not. And this makes other areas of the paper even harder to follow.

Why does the classifier also take R? Additionally, the features are defined as "R union ?".
This is redundant? Also, is training or prediction meant here...?

X_(0) is never defined but is used later when we do X_(pi).

There is a difference in the missingness-process between the observed missingness label \in {0,1} and its associated probability. I had the impression that the authors conflate the two.

**Questions For Authors:**

1. Section 3.4.1 (Unbiased Estimation of the Full Data LOCO Metric) suggests that unbiased estimation is impossible. A conclusive statement reinforcing this would help. What does unbiased refer to?

2. As written above, the distinction between Observed Data LOCO and LOCU is not entirely clear. Providing examples for each scenario would help clarify how the expectation values differ. How do the LOCO measures relate to each other if you have a dataset that contains no missing values? Is LOCO^{OD} = LOCU?

3. If a feature has nearly never been observed, can FMIG still assign it a measurement importance? If yes, how? Is it assumed that the learner has encountered such data points during training? Can we infer a feature importance value from this, or would it simply be a measurement importance? I would expect the latter.

**Relation To Broader Scientific Literature:**

The authors state that this is the first work to evaluate LOCO in the context of missing values. I am also not aware of any other research addressing this topic. The paper cites works from both the feature importance and missing values literature, evaluating the advantages and disadvantages of various approaches in the missing values research field.

**Theoretical Claims:**

I have reviewed the proof of Lemma 4.1 in the appendix. Unfortunately, I am only partially familiar with Robins' g-formula, which prevents me from fully understanding this (fundamental) step.

The definition of the FMIG is motivated by literature in the appendix (A.2). This should be part of the main text. A more detailed derivation of the formulation and construction of the newly introduced measure (FMIG) should be provided, at least in the appendix.

---

> ### Author Rebuttal · Authors · 2025-03-31
>
> We sincerely thank the reviewer for their detailed feedback, which has helped us to improve our paper.
> ## Methods and Evaluation Criteria:
> For clarification on estimation, imputation, and the definition of bias, please refer to our response to reviewer rtga. To address the question about a distinction between the metrics, we have provided a simplified diagnostic scenario involving heart attacks and troponin levels in the introduction. Please find a slightly shorter version below:
> ### Scenario 1: Biological Relationship Analysis (Full Data LOCO)
> A biomedical researcher aims to understand the intrinsic biological relationship between troponin levels, measured via a blood test, and heart attacks, independent of clinical testing protocols. The full data $LOCO$ metric is ideal to assess the explanatory power of troponin values, as it isolates the pure biological relationship, unaffected by real-world data missingness.
>
> ### Scenario 2: Clinical Prediction Model Development (Observed Data LOCO)
>
> A machine learning engineer develops a heart attack prediction model and evaluates the relevance of troponin levels under current real-world missingness patterns. The observed data $LOCO$ metric is used to assess feature importance, ensuring the model aligns with the hospital’s existing testing protocols.
>
> ### Scenario 3: Testing Protocol Optimization (FMIG and LOCU)
>
> After deploying the prediction model, an analyst seeks to optimize troponin testing by evaluating how different testing frequencies affect diagnostic performance. They use FMIG to quantify the benefits of increasing test frequency and LOCU to assess the impact of discontinuing troponin testing entirely.
>
> ## Theoretical Claims:
> We have integrated the relevant appendix sections (A.2) into the main body and expanded the derivations of the FMIG formulation.
> ## Experimental Design and Analysis:
> ### 1. Experiment Design:
> We chose a time-series setting due to its prevalence in medical prediction tasks. Presenting it directly in the main paper would have been overly complex.
> ### 2. Experiments 1 and 2:
> The reviewer mentions that the experiment results confirm what is expected. We agree and want to emphasize that our goal is not to provide novel insights on missingness but to underscore the crucial difference between observed and full data feature importance metrics, challenging current practices that misapply estimators.
> ### 3. Experiment 4:
> Experiment 4 mirrors Experiment 1 but introduces stronger missingness. We've clarified this in Table 4 of the appendix and revised the description in the main text.
>
> The differences in ground truth between Figures 2A and 2B stem from variations in missingness patterns influencing classifier training. Our ground truth reflects feature importance for a classifier trained on the available data which differs between experiments.
> ## Strengths and Weaknesses:
> We thank the reviewer for pointing out errors in our notation. The following adjustments have been made:
>
> We updated the classifier and loss function definitions. The classifier is now $ f_{cl} \colon (\mathbb{R} \cup \lbrace "?" \rbrace )^d \times \lbrace 0,1 \rbrace ^d \to [0,1]^K $, and the loss function is $ L \colon \{0, 1, \ldots, K-1\} \times [0,1]^K \to \mathbb{R} $.
>
> To clarify when a feature is excluded, we define $ X_{-j}, R_{-j}, X_{(1),-j} $ as the reduced dimension observed features, missingness indicators, and ground truth features, and the classifier excluding feature $ j $ is $ f_{cl, -j} \colon (\mathbb{R} \cup \lbrace "?" \rbrace )^{d-1} \times \lbrace 0,1 \rbrace ^{d-1} \to [0,1]^K $.
>
> While we acknowledge the redundancy of $ R $ in the classifier, we kept it to highlight the classifier's dependence on missingness. We now explicitly include $ R$ in all classifier definitions, including the full data case.
>
> Regarding $ X_{(0)} $, we added a reference for readers unfamiliar with potential outcomes: (Rubin, 2005).
>
> Finally, in response to the comment about $ \pi $'s mapping, we corrected the definition of $ \pi$ to represent probabilities. It now maps into $ [0,1]^{2^d}$, rather than $ \lbrace 0,1 \rbrace ^d$.
> ## Questions for Authors:
> ### 1. Distinction Between Metrics:
> If no missingness exists, $LOCO^{OD} = LOCO^{FD}$ since $R = \vec{1}$ and $X = X_{(1)}$. However, $LOCU$ remains generally unidentifiable without knowledge of how leaving feature $j$ unmeasured affects other measurements.
> ### 2. FMIG when Features Are Rarely Observed:
> If a feature is never observed, it is impossible to estimate its measurement impact, violating positivity assumptions. FMIG is based on an odds ratio, so if initial measurement probability is zero, the intervention effect is also zero. This does not mean increased measurement is unhelpful—just that our definition precludes estimation in such cases. When a feature is rarely observed, identification holds, but finite-sample issues can cause non-robust estimates (see Figure 2).

---

> > ### Comment · Reviewer_sya3 · 2025-04-03
> >
> > I thank the authors for their thoughtful rebuttal and appreciate their planned changes. Nonetheless, I still have open issues:
> >
> > 1. Regarding your answer to reviewer sya3: In the proof of the positivity assumption, it must also hold that $\\delta\_j \\cdot \\pi\_j + 1 - \\pi\_j > 0$ (this holds, but it should be written down).
> > Apart from this, I would appreciate it if you could give examples for learners taking R as an input.
> > 2. The examples for $LOCO^{FD}$ and $LOCO^{OD}$ are useful. Still, my above-mentioned point (“The distinction between these perspectives is not entirely clear, leaving some open questions. For instance, both $LOCO^{FD}$ and $LOCO^{OD}$ can be approximated by imputing missing values and differ only in the imputation method used.”) is not fully addressed. The distinction is clear from the application side, but the mathematical definition is still inconclusive insofar as both LOCO estimations allow for imputation.
> > 3. The authors did not comment on my claim that LOCO provides different insights into the data than their newly introduced measure, FMIG. I think this is an important point to mention since users of FMIG could accidentally draw wrong implications otherwise.
> > 4. Experiment 4: The authors write that the difference in ground truth LOCO importance stems from the missingness in the data. In my definition of a ground truth LOCO importance, it would be based on the *true* underlying data-generating function. It should not differ since the features still stem from the same distribution, whether observed or not. If the “ground truth” depends on the underlying data, it must clearly be stated as an approximation that - in this experiment - would distort the comparison.
> > 5. Additionally, for reproducibility, the code must be published.
> >
> > If the authors respond to these aspects and adopt their manuscript accordingly (esp. points 3-5), I would be willing to improve my score.

---

> > > ### Author Response · Authors · 2025-04-08
> > >
> > > Thank you for engaging in the discussion and for your constructive feedback.
> > > Please find our responses to your comments below.
> > >
> > > ## Comment 1:
> > > We agree and have added the additional statement that $\delta_j \cdot \pi_j + 1 - \pi_j > 0$ holds,  for completeness.
> > >
> > > Regarding examples for classifiers taking $R$ as an input, we will add two references regarding the benefit of doing so: [1,2]
> > >
> > > [1] Van Ness, Mike, et al. "The missing indicator method: From low to high dimensions." Proceedings of the 29th ACM SIGKDD Conference on Knowledge Discovery and Data Mining. 2023.
> > > [2] Singh, Janmajay, Masahiro Sato, and Tomoko Ohkuma. "On missingness features in machine learning models for critical care: observational study." JMIR Medical Informatics 9.12 (2021): e25022.
> > >
> > > ## Comment 2:
> > >
> > > We believe that our paper, along with our response to reviewer rtga (which will be incorporated into the manuscript), clearly outlines which imputation methods yield unbiased estimators for the $LOCO^{FD}$ and $LOCO^{OD}$ metrics. However, we agree that a clearer interpretation of these results would enhance the paper, and we provide this below, which we will also include in the revised manuscript.
> > >
> > > When using imputation for the estimation of $LOCO^{FD}$ and $LOCO^{OD}$, the missing values must be imputed in a manner consistent with the classifier’s respective "working conditions."
> > >
> > > For $LOCO^{FD}$, the classifier operates under full data availability. Therefore, missing values must be imputed using samples from the full data distribution. This is achieved by using observed values and imputing missing values according to a density that models the ground truth values of $X_{(1)}$. When done correctly, $LOCO^{FD}$ is evaluated on a dataset that mirrors the original dataset as if no missingness had occurred. In this case, the classifier can leverage the imputed values to potentially improve performance.
> > >
> > > For $LOCO^{OD}$, evaluation occurs under conditions where missingness is present. Since the observed data already reflects the classifier's working conditions, no imputation is required. However, one may opt for impute-then-regress classifiers, where imputation serves purely as a means to handle missing values elegantly. In such cases, the imputation step does not introduce additional information beyond what is already present in the observed input features.
> > >
> > > ## Comment 3:
> > > We assume the reviewer is referring to their comment on Experiment 3.
> > >
> > > We apologize for not addressing this comment directly in our previous response. At the time, we felt it reiterated a point already made in the paper. However, we now recognize that this distinction should be emphasized more clearly for the reader.
> > >
> > > To address this, we have added the following sentence to the discussion section (line 427, right column):
> > >
> > > "FMIG is thus not an alternative to the observed data or full data LOCO metrics, but an additional tool to assess the importance of a feature under a change in measurement probability."
> > >
> > > ## Comment 4:
> > >
> > > We define the feature importance metric as the feature importance given a classifier. This metric accurately represents the true feature importance only if the classifier correctly models the actual probability of the label given the set of features and missingness indicators.
> > >
> > > However, obtaining such a classifier is non-trivial—particularly in the full-data setting—as we discuss in Remark 3.2 (Appendix C) and demonstrate in our experiment in Appendix G.
> > >
> > > In our experiment, the term ground truth refers to the feature importance computed using a (potentially imperfect) classifier. Since any classifier must be learned from the observed data, and we aim to evaluate the metric for the same classifier using different estimation methods (as shown in Figure 2), we necessarily train the classifier on the observed data. Because the observed data varies between experiments, the corresponding classifier—and consequently, the ground truth feature importance estimates—also differ across experiments.
> > >
> > > As expected, the losses are (slightly) higher in Figure 2B because the corresponding observed data contains more missingness, making it more challenging to train a classifier.
> > >
> > > In conclusion, the observed differences do not arise because the ground truth is an approximation, but rather because it is defined based on a classifier that differs between experiments.
> > >
> > > We acknowledge that this distinction is complex and may not be immediately apparent to the reader. To mitigate potential confusion, we have added an explanation in the paper.
> > >
> > >
> > > ## Comment 5:
> > > We apologize for not addressing this in our initial response. The experiments were conducted as part of a broader project on active feature acquisition, which is currently under preparation and scheduled as a software paper in September 2025 as the last publication of my PhD thesis. We hope the reviewer understands that, due to this, we are unable to share the code publicly at this time.

---

### Official Review · Reviewer_rtga · 2025-03-13

**Overall Recommendation:** 4

**Summary:**

This paper introduces a conceptual framework that distinguishes between feature importance methods under missing data: (1) full-data feature importance evaluates each feature's importance if all feature values were present; (2) observed-data feature importance evaluates each feature's importance based on the actual observed data, which include missing features; and (3) feature measurement importance evaluates each feature's importance based on model improvement when the feature is measured with a higher probability.

Empirically, this paper instantiates LOCO (leave-one-covariate-out) for full-data and observed-data feature importance, as well as introduces the FMIG (Feature Measurement Importance Gradient) for feature measurement importance. With synthetic datasets, the authors demonstrate that these three methods offer complementary insights. Finally, the authors demonstrate that violations of the positivity assumption can introduce bias for full-data feature importance.

**Claims And Evidence:**

- In 3.4.2, it is claimed that mean imputation for observed-data feature importance is unbiased. It is unclear what bias means for observed-data feature importance. Hence, it is also unclear why predictive information can lead to bias.

- In 3.4.3, it is strongly claimed that conditional mean imputation introduces bias for both full-data and observed-data feature importance. For observed-data feature importance, I have the same concern as in the first bullet point. For full-data feature importance, given how strong the claim is, although the intuitive explanation makes sense, more solid theoretical or empirical evidence should be presented to support the strong claim.

- For 3.4.3, the authors should discuss whether these biases are also present for methods that consider feature interactions (e.g., the Shapley values). Otherwise, the authors should be clear that their exposition focuses on LOCO.

- Overall, the claims about biases should be strengthened by clearly defining what biases mean for full-data feature importance, observed-data feature importance, and even feature measurement importance. Also, it should be made clear whether those claims are specific to LOCO or applicable to feature importance methods generally.

**Essential References Not Discussed:**

N/A

**Experimental Designs Or Analyses:**

I checked all the experimental designs and analyses, and I found no particular issues.

**Methods And Evaluation Criteria:**

The problem focuses on demonstrating different qualitative insights gained from full-data LOCO, observed-data LOCO, and FMIG. Hence, the qualitative evaluation using synthetic datasets makes sense.

**Other Comments Or Suggestions:**

- Experiment 4 should be presented before Experiment 3. This will complete the analyses for full-data and observed-data feature importance before introducing additional insights from FMIG.

- The full-data classifier doesn't take the missingness indicators as inputs. This should be noted in the subsection labeled **Classifier** in 3.1 for clarity.

- In Equation (5), it is unclear how $\pi_1(R_1 | R_{-0}, X_{-0})$ is defined. I think a special case is needed when $i=1$.

**Other Strengths And Weaknesses:**

**Strengths**

- The formalism that distinguishes full-data vs. observed-data feature importance is clear and empirically useful (as demonstrated by the synthetic experiments).

**Weaknesses**

- The experiments are done for only one set of data-generating and missingness-generating processes. Other generation processes should also be included to make the empirical claims stronger.

- The authors noted that FMIG provides descriptive insights for feature measurement importance. It is unclear how such descriptive insights are useful in practice. Isn't the main point of feature measurement importance to prescribe which features to measure more often?

**Questions For Authors:**

N/A

**Relation To Broader Scientific Literature:**

Although prior methods exist for computing feature importance scores under missing data, this paper provides a clear conceptual framework that distinguishes between the purposes of those methods (i.e, full-data feature importance, observed-data feature importance, vs. feature measurement importance). This conceptual framework can bring clarity when new methods are developed to address feature importance under missing data.

**Theoretical Claims:**

I checked all the proofs.

- In lines 790-800 of the proof for Lemma 4.1, it's only shown that $\pi_{j, \delta_j} = 0$ implies $\pi_{j} = 0$, and that $\pi_{j, \delta_j} = 1$ implies $\pi_{j} = 1$. Does $\pi_{j, \delta_j} \in (0, 1)$ implies $\pi_j > 0$? This is also necessary to prove that positivity holds.

- If the above issue is addressed, then the proof for Corollary E.1 seems correct to me.

---

> ### Author Rebuttal · Authors · 2025-03-31
>
> We appreciate the reviewer's positive feedback and valuable suggestions for improvement. Please find our responses below.
>
> ## Claims and Evidence
>
> ### Definition of Bias
>
> We define $\theta$ as an estimand and $\hat{\theta}$ as its estimator. The bias of the estimator is given by:
> \begin{equation}
>     \text{Bias}(\hat{\theta}) = \mathbb{E}[\hat{\theta}] - \theta.
> \end{equation}
> ### Generalization of Metrics
>
> Our results extend to a broader class of full data feature importance metrics:
> \begin{equation}
>     \theta^{FD} = \mathbb{E} \left[ g( \lbrace f_{cl,s}(X_{(1),s}, R_{s} = \vec{1}) : s \in \mathcal{S} \rbrace, Y ) \right],
> \end{equation}
> where $g$ can be any function and $\mathcal{S}$ is the set of all feature subsets. This definition includes a wide range of metrics, including LOCO and Shapley values. Notably, Shapley values can be expressed as a weighted average of LOCO values across submodels (Verdinelli and Wasserman, 2023).
>
> Additionally, we consider observed data feature importance metrics $\theta^{OD}$ of the same form, with $X_{(1),s}$ and $R_{s} = \vec{1}$ replaced by $X_s$ and $R_s$.
>
> ### Full Data Feature Importance Metrics
>
> As requested, we demonstrate that conditional mean imputation results in biased estimation of full data feature importance metrics. Our analysis is based on the formulation used for the unbiased MI estimator:
> \begin{equation}
>     \theta^{FD} = \sum_{X_{(1)},Y} g( \lbrace f_{cl,s}(X_{(1),s}, R_{s} = \vec{1}) : s \in \mathcal{S} \rbrace , Y ) p(X_{(1)},Y)
>     = \sum_{X_m, X_o, Y, R} g( \lbrace f_{cl,s}(X_{m \cap s}, X_{o \cap s}, R_{s} = \vec{1}) : s \in \mathcal{S} \rbrace, Y ) p(X_m|X_o,Y, R) p(X_o,Y, R),
> \end{equation}
> where $p(X_m|X_o,Y, R)$ represents the imputation density. When the imputation model is learned, one can apply Monte Carlo integration to obtain the unbiased estimator.
>
> Conditional mean imputation, however, simplifies the above expression to:
> \begin{equation}
>     \theta^{FD} \approx \sum_{ X_o, Y, R} g( \lbrace f_{cl,s}( \mathbb{E}[X_{m \cap s}|X_o,R], X_{o \cap s}, R_{s} = \vec{1}) : s \in \mathcal{S}   \rbrace, Y ) p(X_o,Y, R),
> \end{equation}
> which assumes that the expectation operator can be pulled inside the functions $g$ and $f_{cl}$. This assumption is only valid if both functions are linear, which is generally not the case. Consequently, the use of conditional mean imputation introduces bias.
>
> ### Observed Data Feature Importance Metrics
>
> Next, we demonstrate that mean imputation (or a broader class of imputation methods) does not introduce bias for observed data feature importance metrics $\theta^{OD}$. Since $\theta^{OD}$ is defined as a function of the classifier $f_{cl}$, the reported feature importance metric reflects classifier-specific importance rather than a general global measure.
>
> A commonly used class of classifiers, referred to as impute-then-regress classifiers, first impute the missing values and subsequently classify. If the classifier is sufficiently flexible and the dataset is large enough, the choice of imputation method becomes inconsequential, as no new information is introduced. Thus, any classifier that maps $X_s$ and $R_s$ to $Y$ can be used, including those employing mean imputation, without inducing bias.
>
> However, this conclusion holds only if imputation is performed within the classifier itself using only its input features. If conditional mean imputation is applied to the entire dataset before choosing the subset for the classifier, bias arises. Let the imputed features be:  $X'_i =
>         X_i$  if  $R_i = 1$ and $X'_i =  \mathbb{E}[X_i| X_o, R = \vec{1}]$ if $R_i = 0$.
>
> The resulting estimator contains terms $f_{cl,s}(X^\prime_s, R_s)$ and is thus no longer a function of $X_{s}$, and $R_{s}$, but of the whole $X_o$. Consequently, applying conditional mean imputation before classification results in biased estimates.
>
> ## Theoretical Claims:
>
> We added the following to complete the proof for the positivity assumption:
>
> "In settings with $ \pi_j \in (0,1)$, we find
> $ \pi_{j,\delta_j} = \frac{\delta_j \cdot \pi_j}{\delta_j \cdot \pi_j + 1 - \pi_j} > 0
> $
> since $\delta_j \cdot \pi_j$ > 0."
>
> ## Other Strengths and Weaknesses:
>
> The reviewer asked for clarification on FMIG’s practical use and the value of ‘descriptive’ insights. We have updated the text to avoid the ambiguous term ‘descriptive’. FMIG will inform us about what features should be measured more frequently, but doesn’t suggest specific policy interventions.
>
> ## Other Comments or Suggestions:
>
> We've switched experiments 3 and 4 according to the reviewer's request. We also included $R$ in the full data classifier for clarity. We've also clarified that we mean  $R_{-0} \equiv \lbrace \rbrace$.

---

> > ### Comment · Reviewer_rtga · 2025-04-08
> >
> > The authors have addressed the concerns I raised. I encourage the authors to include an exposition about bias w.r.t. to the choice of imputation in the Appendix (or the main text if space permits). I have raised my score from 3 to 4 accordingly.

---

### Decision · Program_Chairs · 2025-05-01

**Decision:**

Accept (poster)

**Comment:**

This paper studies the problem of feature importance when data is not fully observed. Feature importance is a central tool in explainable AI; however, the challenge of handling missing data in the computation of feature importance has been largely overlooked. Therefore, this paper makes a notable contribution by formulating the problem and proposing initial solutions. The reviewers provided suggestions for additional experiments and presentation enhancements, which the authors are encouraged to consider.